# A Corpus Analysis of the Effects of Definiteness and Animacy on Word Order Variation

Hiwa Asadpour

Institute for English and American Studies (IEAS), Department of Linguistics, Goethe University Frankfurt, 60323 Frankfurt am Main, Germany; asadpour@lingua.uni-frankfurt.de

**Abstract:** This article deals with the analysis of word order variation regarding subjects, direct objects, and non-direct object phrases called the "Target" in the corpus of languages of northwestern Iran, viz., Armenian, Mukri Kurdish, and Northeastern Kurdish (Indo-European), Jewish Northeastern Neo-Aramaic (Semitic), and Azeri Turkic (Turkic). The objective is to examine the effects of formal and semantic (in)definiteness in combination with animacy on Target word order variation to find out which one can be a triggering factor.

**Keywords:** formal definiteness; semantic definiteness; animacy; direct object; target; word order

## 1. Introduction

The sample languages in this study include languages that are considered left-branching (i.e., the finite verb appears in the final position as "subject-object-predicate", for example, Iranian, Armenian, and Turkic (cf. Hoffman 1995, 13; Lee 1996, 2; Dum-Tragut 2009; Skjærvø 2009, 94, sct. 5.1; Dryer 2013; Haig and Khan 2019; Bulut 2022, Faghiri et al. 2022)) and right-branching languages (i.e., the finite verb appears in an earlier position as "predicate-subject-object", for instance, Semitic (cf. Lipiński 2001, 500; Haig and Khan 2019, 21)). An exception to such classification is the word order of a specific group of semantic roles called the "Target" (T). Target is a cover term for the semantic roles of the physical Goals of MOTION and CAUSED-MOTION verbs, the metaphorical Goals of SHOW and LOOK verbs, the addressees of verbs of speech, i.e., SAY verbs, the recipients of verbs of transfer, i.e., GIVE verbs, the Resultant States of Change-of-State verbs, and in part, also EXPERIENCERS and BENEFICIARIES.[1] See the examples in Section 2 for an illustration.

The focus languages in this study are low-resource and minority languages of northwestern Iran, which have been in contact for centuries. The sample languages include Mukri Kurdish, Northeastern Kurdish (NEK), Jewish Neo-Aramaic (NENA), Armenian, and Azeri Turkic, all of which are under the superstratum of Persian, the official language of Iran (see Figure 1 below).

"Kurdish" is an umbrella term for several genetically related varieties spoken in the regions of western, northern, and northeastern Iran, northern Iraq, eastern Turkey, eastern Syria, Azerbaijan, Armenia, and Georgia. Large communities also dwell in diasporas in locations such as Europe, North America, and Asia (cf. MacKenzie 1961; Jügel 2014, 2015; Öpengin and Haig 2014; Öpengin 2016; Haig and Öpengin 2018; see Asadpour 2021, 2022a, 2022b for details). Northeastern Neo-Aramaic (NENA) is a Semitic language and is generally used to describe distinct varieties used by Jewish and Christian communities (cf. Khan 2017). Modern Armenian is defined as an independent branch of Indo-European languages and includes two main sub-groups: Western and Eastern Armenian (Asatryan 1962; Dum-Tragut 2009). Finally, Azeri belongs to a southeastern or Oghuz group of the Turkic language family (cf. Lee 1996; Kıral 2001; Bulut 2022).

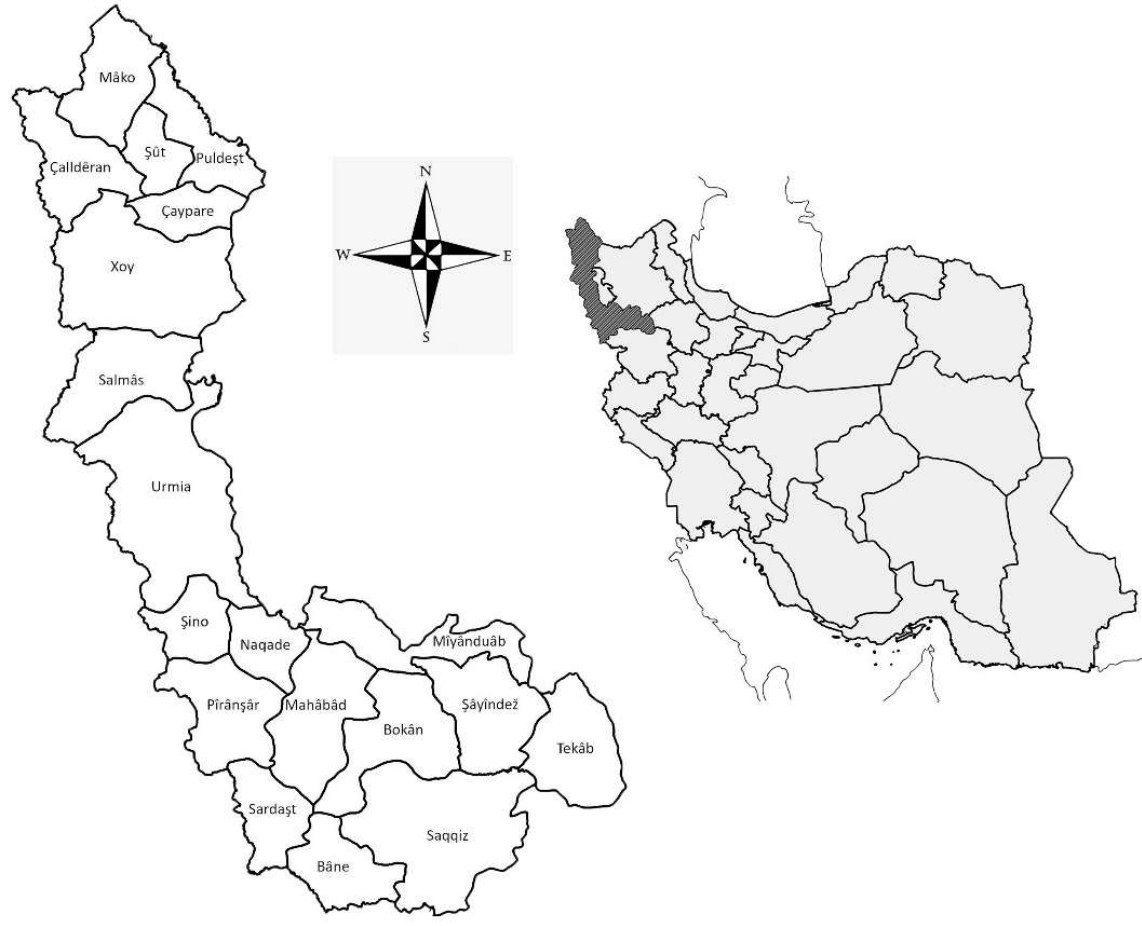

© Asadpour 2021 Left: West Azerbaijan　　　　Right top: Iran/right below: Mukrīyān

**Figure 1.** Western Azerbaijan and its position in northwestern Iran.

This study aimed to conduct a corpus analysis of the variation in word order concerning the definiteness and animacy of Target constituents. The influence of definiteness on word order variation has been widely discussed in the literature (Butler et al. 2010; Vogels and van Bergen 2013). It is generally posited that definite elements tend to occupy earlier syntactic positions due to their higher salience compared to indefinite elements. According to the existing literature, semantically pronominal elements are typically categorized as definite and are, therefore, positioned at the beginning of a clause. These elements are predominantly animate. Likewise, nominal elements marked for definiteness and animate nominal elements are expected to appear early in a clause, while inanimate and indefinite nominal elements are typically found later in a clause (Kittilä 2006; Brunetti 2009; Butler et al. 2010; van Bergen 2011; Vogels and van Bergen 2013 among many others). Among the sample languages, Mukri, NEK, Armenian, and NENA present formal definite markers. Among all of these, Mukri demonstrates a more detailed formal definite marking. While Azeri lacks a distinct formal definite marking system, its inclusion in this study is essential. This is because Azeri, being in contact with other languages in the region under investigation, allows for the exploration of both semantic definiteness and animacy (see Section 3).

The goal of this study is to contribute to areal word order typology. The analysis is based on a comparative examination of the aforementioned languages through a corpus-based approach. The results of this study will offer rich input for providing explanations for word order variation. To conduct the analysis, I used personal field data on Armenian, Azeri Turkic, Mukri, and Northern Kurdish varieties and Christian Neo-Aramaic of northwestern Iran (abbreviated as the TONI corpus). The summary is as follows:

All of the corpora are transcribed, translated, and partially annotated (see Asadpour 2022a for detailed information on the data).

The postverbal placement of Goals in the languages of eastern Anatolia (including Northern Kurdish) received special attention from Haig (2015, 2017, 2022) and Haig and Khan (2019), who spoke in favor of an "areal epicentre" for the "northern Iraq and neighboring regions of western Iran". A typological overview of the Araxes-Iran languages is provided by Stilo (2018), and four Balochi varieties are discussed by Jahani (2018).

Following the contact-induced change explanation by Haig (2015, 2017, 2022) and Haig and Khan (2019), various studies investigated the postverbal placement of Targets in the outlier language varieties of the sample languages in this study, and they offered explanations for Target word order variation, for example, a theoretical account (e.g., Wasow 2022); a multifactorial analysis of Kurdish, Neo-Aramaic, Azeri, and Armenian (cf. Asadpour 2022a, 2022b, 2022c); Middle Iranian (e.g., Jügel 2022); Southern Balochi (e.g., Korn 2022); Iran Turkic (e.g., Bulut 2022); Chulym Turkic (e.g., Lemskaya 2022); NENA (e.g., Noorlander and Molin 2022); and Iraqi Kurdistan Arabic (e.g., Birnstiel 2022).

Cross-linguistically, the word order of dative constructions and the semantic roles of Goals have been investigated in the works of Tomlin (1986), who provided a semantic analysis; Arnold et al. (2000) and Wasow (2002), who offered a discourse-pragmatic explanation; Hawkins (1994, 2004); Gibson (1998); Jaeger and Buz (2018); and Jaeger and Tily (2011), who examined word order variation based on a cognitive and information-theoretic approach (see Asadpour 2022a for a detailed overview of the literature).

In this research, the data were investigated to answer the following questions: What kinds of word orders exist in specific types of definite and indefinite as well as animate and inanimate Targets, and what is the possible relationship between word order, definiteness, and animacy? Does definiteness or animacy or a combination of both play a role in the word order variation in Targets? Furthermore, how are these tendencies represented in the interaction of word order with definiteness and animacy?

## 2. Definiteness

In this paper, I distinguish formal definiteness, i.e., through the formal reflection of determination, and semantic definiteness, i.e., through the property of referentiality. Definiteness and indefiniteness in the sample languages are marked morphologically by the affixation of markers to the nouns. The data will be evaluated for the relationship between formal and semantic definiteness (see Section 4) and Target word order. In addition, the cardinal number "one" is frequently used as an indefinite article (Kurdish *yak/yek*, NENA *xa*, Armenian *mi*, and Azeri *bir*). The cardinal number "one" in Mukri, NEK, and NENA can also be combined with the word *dāna*, literally "grain", and this combination (NENA *xa dank*[2], Kurdish *yek dank/dāna*) expresses indefiniteness as well (see Table 1 below). Here, *dāna* is not a count word, and it is used with all different types of nouns, including animate ones. An overview of the morphological marking is given in Table 1.

**Table 1.** Toni Corpus metadata (Asadpour 2022a).

| Genre | Language | Length (min) | Narrator M-F/Age |
| --- | --- | --- | --- |
| Anecdote | Mukri | 20:15 | M/88, 68 |
| | NEK | 42:36 | F/49, 55, 43, 35 M/65, 57, 38, 32 |
| | C.NENA | 37:54 | F/45, 56, 40 M/34, 49, 30, 57 |
| | Armenian | 18:88 | F/40, 35, 18 M/32, 29, 26 |
| | Azeri | 32:23 | M/65, 58, 46, 39 |

**Table 1.** *Cont.*

| Genre | Language | Length (min) | Narrator M-F/Age |
|---|---|---|---|
| Procedural text | NEK | 9:38 | F/65, 46 M/38 |
| | Armenian | 17:09 | F/40 M/22, 17 |
| | Azeri | 13:02 | M/58 |
| Folk tale | Mukri | 07:09 | M/89 |
| | NEK | 51:08 | F/89, 78, 74, 69, 57 |
| | Azeri | 17:13 | M/66 |
| Real-life story | Mukri | 17:93 | F/87 M/88, 68 |
| | NEK | 51:05 | F/71, 64 M/58 |
| | C.NENA | 40:32 | F/61, 54 M/47, 58, 59 |
| | Armenian | 07:35 | F/28 M/35 |
| Mixed | Mukri | 41:05 | M/76 |
| Five genres | | 423:83 mins | 55 narrators Age range 17–89 |

Table 2 shows an overview of formal definite marking in the sample languages. Below, I will give examples of formal marking in the sample languages for illustration.

**Table 2.** Morphological marking of (in)definiteness (Asadpour 2022a).

| | Mukri sg. | Mukri pl. | NEK sg. | NEK pl. | NENA sg. | NENA pl. | Armenian sg. | Armenian pl. | Azeri sg. | Azeri pl. |
|---|---|---|---|---|---|---|---|---|---|---|
| DEF | *-a/-aka* | *-akān* | *-ak(a)* | - | *(il-/-la)/-aka* | - | *-ə/-n* | - | - | - |
| -DEF | *-ak* | *-ekān/-ānek* | *-ek* | - | - | - | - | - | - | - |
| Bare | *-∅* | *-ān* | *-∅* | *-ān* | *-∅* | *-e* | *-∅* | - | - | *-lār* |
| | | | | *-en* | | *-a* | | *(n)er* | *∅* | |
| | | | | | | *-ta* | | | | |
| | | | | | | *-la* | | | | |
| | | | | | | *-lta* | | | | |
| | | | | | | *-ane* | | | | |
| | | | | | | *-awe* | | | | |
| "one" | *yak/yek* | - | *yak/yek* | - | *xa* | - | *mi* | - | *bir* | - |

Below, I will present several examples to illustrate definite marking in the sample languages.

(1) Singular –DEF *yak* and DEF *aka* (Mukri, ÖM 2016: 197, NZ.187 cited in Asadpour 2022a)

| | | | S | P | P | T | V |
|---|---|---|---|---|---|---|---|
| *amn=īš* | *nānawā-**yak**=mān* | *habū* | *xom* | *da* | *bar* | *nānawā-y-**aka**-y* | *kərd* |
| 1SG=ADD | bakery-**INDF**=PC.1PL.POSS | have.PST | self.1SG | at | on | bakery-GLID-**DEF**-OBL | do.PST |

"As for me, (we) had a bakery. (I) put myself under the bakery [*lit.* I laid with the bakery].:

In example (1), the object is marked with the indefinite marker *-yak*, whereas the Target is marked with the definite suffix *-aka* and is also flagged by a combination of grammatical and lexical prepositions and an oblique case.

(2) Singular DEF *-a* (Mukri, ÖM 2016: 205, ČQ.139 cited in Asadpour 2022a)

| S | O | V=P | | T |
|---|---|---|---|---|
| *pādšā* | *daz* | *bərd=a* | *sar=ī* | *bāza-**a**-y* |
| king | hand | bring.PST=to | head=EZ | falcon-**DEF**-OBL |

"The king got a hold of the falcon's head.
[*lit.* The king put his hand on the head of the falcon]".

In example (2), the object is unmarked, and the Target is marked by the definite suffix *-a*. The definite marker *-a* is also introduced for the masculine gender as an oblique and as a definite article. It can also have the function of a demonstrative in the form of *=a*, which is cliticized to a nominal element. See example (3) below for an illustration of *-a* as a definite suffix:

(3) *-a* as a definite suffix (Mukri ÖM 2016: 270, ŽB.41b cited in Asadpour 2022a)[3]

| | S | P | T | V |
|---|---|---|---|---|
| *kut=ī,* | *kuř-a* | *wa* | *peš=əm* | *kawət* |
| say.PST=PC.3SG.AGR | boy-**DEF** | to | front=PC.1SG | fall.PST |

"(He) said, the boy fell down in front of me."

In example (3), *kuř* ("boy") is the subject of the sentence, and it is marked by the suffix *-a*. The referent is given and familiar, and it refers to someone who forms part of the background information shared between the speaker and the hearer. Out of context, in a conversational dialogue where both the speaker and the hearer see someone and the speaker points to the person by using the *-a* suffix, this clitic most likely has a demonstrative rather than definite function. In this case, the extra marking by the suffix *-a* is for the purpose of highlighting the specific referent (see Asadpour 2022a, sct. 5.7).

(4) Singular DEF *-ak* (NEK, TONI, SM_63 cited in Asadpour 2022a)

| | O | | P | T | P | |
|---|---|---|---|---|---|---|
| *du* | *toq=e* | *sarī-y=ā* | *we* | *kəç-**ak**-e* | *re* | *īnāndəbū* |
| two | scarf=EZ.M | head-GLID=PL | that.OBL.SG.F | girl-**DEF**-OBL.SG.F | POSTP | bring.PPF |

"(She) brought two head scarves to the girl."

In example (4), the object is unmarked, and the Target is marked by a definite suffix *-ək* and is flagged by a circumposition.

(5) Singular -DEF *-ek* (NEK, TONI, MD_62 cited in Asadpour 2022a)

| | V=P | T |
|---|---|---|
| *ku* | *bə-š-t=a* | *cə-y-**ek**-e* |
| that | SBJV.go.PRS-3SG=to | place-GLID-**INDF**-OBL.SG.F |

"[. . .] that (he) goes to somewhere."

In example (5), the Target is marked with the indefinite suffix *-ek*. In the TONI corpus, NEK has the definite article *-ak(a)*, while Haig and Öpengin (2018, 16) claim that in Northern Kurdish, if a constituent is not morphologically marked with the indefinite suffix, it is considered to be either definite or generic, depending on the context. Furthermore, Gündoğdu (2018, 53) claims that in Kurmanjī, and especially in Muš Kurmanji Kurdish, there is no definite article. In the next examples, I illustrate the definite and indefinite marking of attested tokens in NENA, Azeri, and Armenian.

(6) Singular -DEF *xa-* (NENA (Khan 2008, 414, E87C cited in Asadpour 2022a)).

| O | | V | T |
|---|---|---|---|
| ***xá-danka*** | *pardá* | *yasr-í-wa* | *m-gudà* |
| **one-grain** | curtain | tie.HAB-3PL-PST | PREP-wall |

"(They) would draw a curtain over the wall."

Since, in NENA, Targets are not attested for marking by definite markers, I give the example of a direct object marked with the indefinite *xa-danka* marker. Similarly,

Azeri also presents indefinite marking by the element *bir* in Target constructions (see example (7) below).

(7) Singular -DEF *bir* (Azeri TONI cited in Asadpour 2022a)

|  | O |  |  | V | T |
|---|---|---|---|---|---|
| ***bir-dānā*** | *kītāb* | *āl-dı-m* | *ver-dī-m* | ***bir*** | *kas-a* |
| **one-grain** | book | buy-PST-1SG | give-PST-1SG | **one** | someone-DAT |

"(I) bought a book and gave (it) to someone."

In Armenian, similar to Mukri and NEK, nominal elements can be marked with definite suffixes (see examples (8), (9), (10), and (11) below).

(8) Singular DEF -*n* (Armenian TONI, 3-1.25 cited in Asadpour 2022a)

|  | T |  | V |  |  |
|---|---|---|---|---|---|
| *ter* | *Bagrat-**i-n*** | *el* | *asa-m* | *vor* | *ari…* |
| father | Bagrat-**DAT-DEF** | also | say.PST-1SG | that | come.IMP.SG… |

"(I) also told [lit. said] Father Bagrat to come."

(9) Singular DEF -*ə* (Armenian, TONI, 5-2.14m/n cited in Asadpour 2022a)

| V |  |  | T |  |
|---|---|---|---|---|
| *gnaṣ-i* | *ira* |  | *łek-i* | *hetew-**ə*** |
| go.PST-1SG | his |  | steering.wheel-DAT | behind-**DEF** |

"(I) went behind his steering wheel."

(10) Singular -DEF *mi* (Armenian, TONI, 5-2.2a cited in Asadpour 2022a)

| V | T |  |  |  |
|---|---|---|---|---|
| *gnaṣ-inkʿ* | ***mi*** | *hat* | *pʿołoṣ-i* | *ners* |
| go.PST-1PL | **one** | item | street-DAT | inside |

"(We) went to [lit. inside] a street."

Above, I gave examples for singular forms. Below, I show examples of elements with definiteness in the plural form. All examined languages express (in)definite plurality through the ending Mukri -*ān*; NEK -*en*, NENA -*e*, Armenian -*(n)er*, and Azeri -*lār* (see Table 2 above).

(11) Plural DEF -*ak-ān* (Mukri, ÖM 2016: 226, MK.223 cited in Asadpour 2022a)

| V |  | P |  | T |  |
|---|---|---|---|---|---|
| *da-č-et=awa* |  | *kən* | *dəz-**ak-ān*** | *čəl* | *nafar-aka-y* |
| IPFV-go.PRS-3SG=POSTV |  | close | thief-**DEF-PL** | forty | individual-DEF-OBL |

"(He) goes close to the thieves, the forty men."

(12) Plural DEF -*k-en* (NEK, TONI, SM_96 cited in Asadpour 2022b)

|  | S |  | V=P | P |  | T |
|---|---|---|---|---|---|---|
| *čəra* | *motorkəlāzā* | *kur-ək* | *na-hāt=ā* | *pešə-y=ā* | *bərā-**k-en*** | *kəç-ək-e* |
| why | motorbike | boy-DEF | NEG-come.PST=to | front-GLID=EZ.F | brother-**DEF-PL.EZ** | girl-DEF-OBL.SG.F |

"Suddenly the motorbike of the boy came to the front of the brothers of the girl."

(13) Plural DEF -*ner-in* (Armenian, Dum-Tragut 2009, 67 cited in Asadpour 2022a)

|  | S | V |  |  | O |  |
|---|---|---|---|---|---|---|
| *es* | *tesn-um* | *em* | *ays* | *erekʿ* | *ałjik-**ner-i-n*** |
| I | see-PTCP.PRS | COP.1SG | this | three | girl-**PL**-DAT-**the** |

"I see these three girls."

In Armenian, no Targets with plural and definite suffixes are attested. Finally, Targets and other elements can be unmarked, i.e., without any definite marking:

(14) Unmarked Target (Mukri ÖM 2016: 254, ČN.118 cited in Asadpour 2022a)

| O | P | **T** | V |
|---|---|---|---|
| *məndāł-a=yān* | *da* | ***sundūq-e*** | *hāwīšt* |
| child-DEF=PC.3PL.AGR | into | **COFFER-OBL** | throw.PST |

"(They) threw the child **into the coffer**."

(15) Unmarked Target (NEK, TONI, KP_159 cited in Asadpour 2022a)

| V | T |
|---|---|
| *hat-in* | *mal-e* |
| come.PST.3PL | home-OBL.SG.F |

"(They) came **home**."

(16) Unmarked Target (NENA, Khan 2008, 418, F101B cited in Asadpour 2022a)

| V | P-T |
|---|---|
| *zəllu* | *g-komsèr* |
| go.PST.3PL | OBL-police_station |

"(They) went to **the police station**."

(17) Unmarked Target (Armenian, TONI, 5-1.28 cited in Asadpour 2022a)

| | V | T |
|---|---|---|
| *heto,* | *gnac̣inkʽ* | *Iran* |
| then | go.PST.1PL | **Iran** |

"Then (we) went to **Iran**."

(18) Unmarked Target (Azeri, kıral, 2001: 142, T2/4 cited in Asadpour 2022a)

| | | | O | | | V |
|---|---|---|---|---|---|---|
| *va* | *har* | *na* | *da* | *kī* | *yāz-ır-dī-∅* | *āpār-ır-dı-∅* |
| and | every | what | also | that | write-IPFV-PST-3SG | take-IPFV-PST-3SG |

| T | | | | | |
|---|---|---|---|---|---|
| ***ruznāmī- ya*** | *kī* | *cāp* | *elī-ya-lar* | *cāp* | *ela-m-īr-dī-lar* |
| **newspaper-DAT** | that | publishing | do-OPT-3PL | publishing | do-NEG-IPFV-PST-3PL |

"and whatever (he) was writing would be taken to **the newspaper** so that they publish it but they would not publish it."

## 3. Definiteness, Animacy, and Word Order Variation in Cross-Linguistic Studies

Vogels and van Bergen (2013, 2–3) propose that definiteness serves as an indicator of a referent's accessibility within a discourse, a concept termed "discourse accessibility", while they regard animacy as an inherent property of concepts, which they term "inherent accessibility". Their perspective posits that animacy significantly affects the accessibility of a referent, thereby influencing the choice of word order in Dutch. Specifically, they predict that definite and animate subjects, representing highly accessible referents, tend to be favored in the preverbal position. In contrast, inanimate and indefinite subjects, denoting less accessible or "non-referential (bare)" referents, exhibit a reduced preference for the preverbal position. Vogels and Bergen argue that the degree of accessibility may impact the predictability effect of the "Subject First preference" rule, with more accessible referents being stronger competitors for this rule (Vogels and van Bergen, 1). Consequently, they conclude that their findings support a probabilistic approach to the study of syntactic variation. The results of their study align with broader cross-linguistic investigations into word order, where highly accessible referents, including those in the case of NEK, a consistently postverbal language, are typically found in preverbal positions (see Sections 4 and 5).

In an experimental study by Butler et al. (2010), it was demonstrated that definite, human, and animate arguments tend to occur in preverbal positions, while indefinite referents are more commonly found postverbally. Additionally, when both the agent and patient are animate and human, the patient is more likely to be fronted. If the agents are inanimate, the fronting of human patients becomes even more pronounced. Tonhauser (2003) explored the syntactic and semantic factors affecting focus constructions, as well as the impact of definiteness and animacy in Yucatec Maya. Her findings indicate that, in this language variety, full nominal phrases typically appear postverbally, with the preverbal position reserved specifically for definite animate referents that are under focus.

Kittilä (2006, 12–21) argues that the animacy strategy also influences the marking of referents in ditransitive constructions. He provides examples from various languages

to illustrate that animate "Themes" and "Recipients" are marked similarly to animate "Patients," while inanimate "Themes" are marked similarly to inanimate "Patients." Kittilä further suggests that animate referents often exhibit a high degree of definiteness or topicality (Kittilä 2006, 18), although this distinction is not always straightforward.

Similar to the animacy hierarchy, the concept of (in)definiteness, as briefly explained by Kittilä, can be related to established approaches that identify universal tendencies concerning definiteness. For instance, the information structure and definitizing account given by Givón (1979, 1984a, 1984b, 1993, 2001), Dominance Theory put forward by Erteschik-Shir (1979), and information saliency described by Siewierska (1988) all make the general claim that definite referents tend to appear early in a sentence, while indefinite referents are typically positioned later in a clause. Each of these principles is briefly outlined below.

By combining the above three principles, a connection emerges between the verb type, animacy, definiteness, and parts of speech (PoS). In the TONI corpus, new information is usually introduced through an indefinite expression, while given information is referred to using a definite expression, often manifesting as a definite nominal or pronominal referent, and, in the case of the TONI corpus, it can also involve a bound pronoun. Given and definite referents are also referred to as anaphoric elements (see Brown and Yule 1983, 171). Consequently, a correlation between definiteness and givenness, on the one hand, and indefiniteness and new information, on the other, emerges, along with the selection of the part-of-speech type. Nevertheless, the TONI corpus indicates that this is not a strict rule, and given or new information is not always synonymous with definiteness or indefiniteness. As Chafe (1976, 42) notes, it is possible for a definite element to be implicit but retrievable in terms of addressees' consciousness, implying that a new referent can be discourse-new while simultaneously representing old information for the speaker and hearer (see Asadpour 2022a). In the upcoming sections, a detailed examination of corpus-based overview investigations into word order variation will be provided as it pertains to the concepts of definiteness and animacy.

## 4. Corpus Analysis of Formal Definiteness

For different parts of speech, the following coding is used. Nominal elements can be marked by a(n) (in)definite marker, or they can be unmarked. Pronominal elements, as well as bound pronouns, are considered unmarked. Since the pronominal and bound pronoun elements are given information, they are coded as unmarked definite unless they indicate an indefinite element. Table 3 below offers the placement of constituents according to their definiteness on subjects, direct objects, and Targets.

**Table 3.** Formal definiteness marking and word order (X = %).

|    |          |       | Mukri | NENA | Azeri | Armenian | NEK |
|----|----------|-------|-------|------|-------|----------|-----|
| TV | Marked   | DEF   | 40    | 6    | –     | 17       | 6   |
|    |          | -DEF  | 33    | 76   | –     | 7        | 56  |
| VT |          | DEF   | 15    | 0    | –     | 72       | 24  |
|    |          | -DEF  | 11    | 18   | –     | 3        | 14  |
| TV | Unmarked | DEF   | 65    | 39   | 33    | 25       | 13  |
|    |          | -DEF  | 9     | 4    | 5     | 3        | 2   |
| VT |          | DEF   | 23    | 55   | 58    | 64       | 77  |
|    |          | -DEF  | 3     | 2    | 4     | 7        | 8   |

Table 3 shows the frequency of different word orders (TV: Target-Verb; VT: Verb-Target) for marked and unmarked Targets (Mukri, NENA, Azeri, Armenian, NEK) in terms of definiteness (DEF, -DEF). All sample languages demonstrate a tendency for unmarked constituents. Mukri illustrates a fairly equal distribution of formal definite marking in both positions. NENA presents an obvious tendency for the indefinite marking of constituents in the preverbal position. Azeri displays no definite marking, and unmarked constituents are mostly definite regardless of their position. NEK shows a preference for definite marking in

the postverbal position and indefinite marking in the preverbal position. There also seems to be a tendency for unmarked definite arguments to appear postverbally in the sample languages, but there is no real tendency for the unmarked constituent in Mukri.

The research question can be answered by looking at the patterns and trends in the table, as well as performing some tests of significance to compare the proportions of word orders across the variables. To answer the question, the kinds of word orders in specific types of definite and indefinite Targets are detailed in the following paragraphs.

For marked Targets, TV is more common than VT for both definite and indefinite Targets, except for in Armenian, where VT is more common for definite Targets. This suggests that word order variation for marked Targets is influenced by language-specific factors rather than definiteness.

For unmarked Targets, TV is more common than VT for definite Targets, while VT is more common than TV for indefinite Targets. This suggests that word order variation for unmarked Targets is influenced by definiteness rather than language-specific factors.

For both marked and unmarked Targets, there is a significant difference in the proportions of TV and VT across definiteness (chi-square = 181.4, $p < 0.001$). This means that definiteness affects word order variation for both marked and unmarked Targets.

In the next passages, I will show that animacy demonstrates an influence on Target PoS, especially regarding noun phrase placement, and it is necessary to separate the constituents into subjects, objects, and Targets and to analyze the effects of definite marking on each of them separately. This helps to gain a clearer picture of the influence of definiteness on Target word order. For this purpose, it is also important to pair the features, looking at animacy and definiteness together for the different types of constituents. This will help to better examine the data.

Table 4 shows the frequency of different word orders (TV: Target-Verb; VT: Verb-Target) for marked and unmarked Targets (Mukri, NENA, Azeri, Armenian, NEK) in terms of definiteness (DEF, -DEF) and the definiteness of the subject (S) and object (O) constituents. For marked Targets, there is no clear relationship between the word order and the definiteness of the subject or object constituents. The proportions of TV and VT do not vary much across different combinations of subject and object definiteness. This suggests that word order variation for marked Targets is not influenced by the definiteness of other constituents.

**Table 4.** Definite marking of constituents in pre- and postverbal Targets (X = n) (DEF = definite; -DEF = indefinite; U.DEF = unmarked definite; U.-DEF = unmarked indefinite, X = n).

| | | Mukri | | | NENA | | | Azeri | | | Armenian | | | NEK | | |
|---|---|---|---|---|---|---|---|---|---|---|---|---|---|---|---|---|
| | | **S** | **O** | **T** | **S** | **O** | **T** | **S** | **O** | **T** | **S** | **O** | **T** | **S** | **O** | **T** |
| **TV** | DEF | 11 | 37 | 25 | 1 | 0 | 0 | – | – | – | 2 | 1 | 2 | 5 | 1 | 3 |
| | -DEF | 12 | 37 | 12 | 1 | 10 | 2 | – | – | – | 0 | 0 | 2 | 0 | 6 | 2 |
| | U. DEF | 460 | 249 | 403 | 105 | 42 | 101 | 87 | 51 | 68 | 51 | 29 | 37 | 75 | 35 | 72 |
| | U. -DEF | 21 | 72 | 64 | 4 | 11 | 8 | 4 | 3 | 23 | 1 | 3 | 13 | 3 | 11 | 6 |
| **VT** | DEF | 5 | 9 | 14 | 0 | 0 | 0 | – | – | – | 7 | 4 | 10 | 19 | 5 | 11 |
| | -DEF | 7 | 3 | 11 | 0 | 3 | 0 | – | – | – | 0 | 0 | 1 | 10 | 1 | 9 |
| | U. DEF | 186 | 55 | 142 | 143 | 65 | 138 | 168 | 30 | 164 | 145 | 35 | 117 | 292 | 112 | 259 |
| | U. -DEF | 6 | 4 | 37 | 2 | 7 | 6 | 3 | 12 | 7 | 3 | 3 | 27 | 8 | 10 | 50 |

In considering Targets in pre- and postverbal positions, there is no clear pattern of the definiteness distribution in Mukri because formal marking occurs in both positions for all three constituents. This tendency is also the same for unmarked forms. In NENA, indefinite marking shows a clear preference for the preverbal position with all constituents. Unmarked constituents in NENA display no placement sensitivity. Azeri presents no definite marking, and unmarked Targets exhibit a tendency for preverbal unmarked indefinite and postverbal unmarked definite. In Armenian, most of the formal definite marking

occurs postverbally for all constituents and less so in the preverbal position. Unmarked Targets reveal no preference for either position. In NEK, the formal marking of definiteness displays a tendency for the postverbal position, and unmarked forms are neutral in terms of preference. Among various PoS, objects and Targets are mostly marked with (in)definite articles and fewer subjects in Mukri; objects in NENA; more Targets and fewer subjects in Armenian; and more subjects and Targets and fewer objects in NEK. For both marked and unmarked Targets, there is a significant interaction effect between the word order, Target definiteness, and subject or object definiteness (chi-square = 108.9, $p < 0.001$). This means that word order variation depends on the combination of Target definiteness and subject or object definiteness.

For a clearer idea of what is happening for Targets in various positions, it is necessary to separate the Targets and examine them in relation to animacy. Table 4 demonstrates the realization of Targets in terms of definiteness marking.

In Table 5, Targets that are marked with a definite article are in the preverbal position in Mukri for both human (3%) and inanimate (4%) and have a lower tendency to be in the postverbal position for human (2%) vs. inanimate (2%). The indefinite Targets are fairly divided between pre- and postverbal positions. In NENA, indefinite Targets with formal marking are in the preverbal position (1%), and definite Targets are in the postverbal position (1%). Azeri illustrates no definite marking. In Armenian, the definite marking of human Targets presents a preference for the postverbal position among inanimate entities (6%). Finally, NEK has a tendency for postverbal Targets to be marked with a definite article (7% vs. 2% in the preverbal position). In all of these languages, the strongest trend is for unmarked Targets. Unmarked human Targets are more often located before the verb: for example, in Mukri, H = 84% occurred preverbally, while H = 8% of tokens occurred postverbally. On the other hand, inanimate unmarked Targets are located postverbally: for example, I = 52%, whereas I = 33% of tokens are preverbal. Equally preferred for both positions in NENA are unmarked human Targets (preverbal = 52% vs. postverbal = 43%), as well as postverbal preferences for inanimate Targets (79% vs. 18% in the preverbal position). The trend becomes clearer in Azeri, Armenian, and NEK, which show a preference for postverbal unmarked definite Targets.

**Table 5.** Targets' placement, (in)definiteness marking, and animacy (H = human; A = animate; non-human; I = inanimate; X = %).

|    |        | Mukri |    |     | NENA |    |     | Azeri |     |     | Armenian |     |     | NEK |    |     |
|----|--------|-------|----|-----|------|----|-----|-------|-----|-----|----------|-----|-----|-----|----|-----|
|    |        | H | A | I | H | A | I | H | A | I | H | A | I | H | A | I |
| TV | DEF    | 3 | 0 | 4 | 0 | 0 | 0 | – | – | – | 2 | 0 | 1 | 2 | 0 | 0 |
|    | -DEF   | 1 | 0 | 3 | 1 | 0 | 0 | – | – | – | 0 | 0 | 1 | 0 | 0 | 1 |
|    | U. DEF | 84 | 78 | 28 | 52 | 0 | 24 | 55 | 100 | 18 | 46 | 0 | 6 | 38 | 40 | 5 |
|    | U. -DEF | 2 | 0 | 17 | 2 | 0 | 4 | 3 | 0 | 1 | 8 | 0 | 6 | 0 | 0 | 2 |
| VT | DEF    | 2 | 0 | 2 | 0 | 0 | 1 | – | – | – | 2 | 0 | 6 | 7 | 0 | 0 |
|    | -DEF   | 0 | 11 | 3 | 0 | 0 | 0 | – | – | – | 0 | 0 | 1 | 0 | 0 | 3 |
|    | U. DEF | 8 | 11 | 33 | 43 | 50 | 67 | 42 | 0 | 79 | 35 | 100 | 65 | 49 | 50 | 71 |
|    | U. -DEF | 1 | 0 | 10 | 1 | 50 | 4 | 0 | 0 | 2 | 8 | 0 | 15 | 3 | 10 | 17 |
|    | n      | 371 | 9 | 338 | 141 | 2 | 113 | 104 | 2 | 156 | 63 | 1 | 144 | 136 | 10 | 264 |

As shown above, definiteness affects word order variation for unmarked Targets, while animacy affects word order variation for marked Targets. However, there is no clear interaction effect between definiteness and animacy on word order variation. For marked Targets, there is a significant difference in the proportions of TV and VT across animacy (chi-square = 10.8, $p = 0.01$) but not across definiteness (chi-square = 0.01, $p = 0.92$). This means that marked Targets tend to have different word orders depending on whether they are human, animate non-human, or inanimate, but not depending on whether they are definite or indefinite.

For unmarked Targets, there is a significant difference in the proportions of TV and VT across definiteness (chi-square = 113.64, $p < 0.001$) but not across animacy (chi-square = 0.36, $p = 0.55$). This means that unmarked Targets tend to have different word orders depending on whether they are definite or indefinite, but not depending on whether they are human, animate non-human, or inanimate.

For human Targets, there is a significant difference in the proportions of TV and VT across marking (chi-square = 28.69, $p < 0.001$) but not across definiteness (chi-square = 1.59, $p = 0.21$). This means that human Targets tend to have different word orders depending on whether they are marked or unmarked, but not depending on whether they are definite or indefinite.

For animate non-human Targets, there is a significant difference in the proportions of TV and VT across marking (chi-square = 38.64, $p < 0.001$) but not across definiteness (chi-square = 0.01, $p = 0.94$). This means that animate non-human Targets tend to have different word orders depending on whether they are marked or unmarked, but not depending on whether they are definite or indefinite.

For inanimate Targets, there is a significant difference in the proportions of TV and VT across marking (chi-square = 38.64, $p < 0.001$) and across definiteness (chi-square = 113.64, $p < 0.001$). This means that inanimate Targets tend to have different word orders depending on whether they are marked or unmarked and whether they are definite or indefinite.

At the current analytical level, as stated above, the categories are broad enough, and this leads to results of their own. Some finer distinctions that exist in the data, however, may still have been missed. Therefore, in order to give a finer analysis, I have paired the features of the (in)definiteness and animacy of Targets for each language, and I will look at the results for each constituent: subject, object, and Target.

For a closer look at the overt realization of subjects, objects, and Targets, I cross-paired animacy and definiteness for each language. Tables 6–10 below determine the data for subjects, objects, and Targets and their definite marking and animacy in the sample languages. The categories begin with human (h), animate (a), and inanimate (i) definite (DN), followed by human, animate and inanimate indefinite (IN), unmarked definite (UD), and unmarked indefinite (UI).

**Table 6.** Formal definiteness and animacy among overt constituents in Mukri (X = %).

| | **Mukri** | **hDN** | **aDN** | **iDN** | **hIN** | **aIN** | **iIN** | **hUD** | **aUD** | **iUD** | **hUI** | **aUI** | **iUI** |
|---|---|---|---|---|---|---|---|---|---|---|---|---|---|
| Subject | TV | 37 | 60 | 3 | 67 | 75 | 12 | 49 | 55 | 3 | 67 | 20 | 6 |
| (N = 222) | VT | 0 | 0 | 0 | 0 | 0 | 0 | 1 | 3 | 0 | 0 | 0 | 0 |
| Object | TV | 26 | 40 | 94 | 33 | 25 | 62 | 10 | 23 | 33 | 17 | 70 | 48 |
| (N = 317) | VT | 0 | 0 | 0 | 0 | 0 | 0 | 0 | 0 | 7 | 0 | 0 | 0 |
| Target | TV | 9 | 0 | 0 | 0 | 0 | 5 | 15 | 8 | 4 | 0 | 0 | 6 |
| (N = 395) | VT | 29 | 0 | 3 | 0 | 0 | 21 | 25 | 13 | 54 | 17 | 10 | 41 |
| | n | 35 | 5 | 37 | 15 | 4 | 42 | 262 | 40 | 349 | 24 | 10 | 109 |

**Table 7.** Definiteness and animacy among overt constituents in NENA (X = %).

| | **NENA** | **hDN** | **aDN** | **iDN** | **hIN** | **aIN** | **iIN** | **hUD** | **aUD** | **iUD** | **hUI** | **aUI** | **iUI** |
|---|---|---|---|---|---|---|---|---|---|---|---|---|---|
| Subject | TV | 100 | 0 | 0 | 0 | 0 | 9 | 35 | 80 | 3 | 38 | 0 | 0 |
| (N = 100) | VT | 0 | 0 | 0 | 0 | 0 | 0 | 1 | 0 | 0 | 8 | 0 | 0 |
| Object | TV | 0 | 0 | 0 | 33 | 0 | 91 | 6 | 0 | 17 | 23 | 0 | 78 |
| (N = 122) | VT | 0 | 0 | 0 | 0 | 0 | 0 | 0 | 0 | 25 | 0 | 0 | 0 |
| Target | TV | 0 | 0 | 0 | 67 | 0 | 0 | 32 | 0 | 14 | 23 | 0 | 0 |
| (N = 256) | VT | 0 | 0 | 100 | 0 | 0 | 0 | 27 | 20 | 40 | 8 | 100 | 22 |
| | n | 1 | 0 | 1 | 3 | 0 | 11 | 230 | 5 | 189 | 13 | 1 | 18 |

**Table 8.** Definiteness and animacy among overt constituents in Azeri (X = %).

|  | Azeri | hDN | aDN | iDN | hIN | aIN | iIN | hUD | aUD | iUD | hUI | aUI | iUI |
|---|---|---|---|---|---|---|---|---|---|---|---|---|---|
| Subject | TV | – | – | – | – | – | – | 45 | 0 | 1 | 36 | 0 | 0 |
| (N = 95) | VT | – | – | – | – | – | – | 1 | 0 | 0 | 7 | 0 | 0 |
| Object | TV | – | – | – | – | – | – | 3 | 0 | 15 | 14 | 0 | 34 |
| (N = 91) | VT | – | – | – | – | – | – | 1 | 0 | 19 | 0 | 0 | 0 |
| Target | TV | – | – | – | – | – | – | 28 | 100 | 6 | 43 | 0 | 45 |
| (N = 262) | VT | – | – | – | – | – | – | 23 | 0 | 59 | 0 | 0 | 21 |

**Table 9.** Definiteness and animacy among overt constituents in Armenian (X = %).

|  | Armenian | hDN | aDN | iDN | hIN | aIN | iIN | hUD | aUD | iUD | hUI | aUI | iUI |
|---|---|---|---|---|---|---|---|---|---|---|---|---|---|
| Subject | TV | 38 | 100 | 14 | 0 | 0 | 0 | 32 | 0 | 1 | 23 | 100 | 0 |
| (N = 40) | VT | 13 | 0 | 0 | 100 | 0 | 0 | 0 | 0 | 0 | 0 | 0 | 0 |
| Object | TV | 13 | 0 | 14 | 0 | 0 | 0 | 6 | 0 | 13 | 0 | 0 | 14 |
| (N = 54) | VT | 13 | 0 | 0 | 0 | 0 | 0 | 1 | 0 | 14 | 0 | 0 | 5 |
| Target | TV | 13 | 0 | 7 | 0 | 0 | 67 | 35 | 0 | 6 | 38 | 0 | 22 |
| (N = 209) | VT | 13 | 0 | 64 | 0 | 0 | 33 | 26 | 100 | 67 | 38 | 0 | 59 |
|  | n | 8 | 1 | 14 | 1 | 0 | 3 | 84 | 1 | 139 | 13 | 1 | 37 |

**Table 10.** Definiteness and animacy among overt constituents in NEK (X = %).

|  | NEK | hDN | aDN | iDN | hIN | aIN | iIN | hUD | aUD | iUD | hUI | aUI | iUI |
|---|---|---|---|---|---|---|---|---|---|---|---|---|---|
| Subject | TV | 61 | 0 | 14 | 100 | 0 | 12 | 46 | 25 | 6 | 57 | 0 | 4 |
| (N = 197) | VT | 0 | 0 | 0 | 0 | 0 | 0 | 0 | 0 | 0 | 0 | 0 | 0 |
| Object | TV | 5 | 100 | 43 | 0 | 0 | 29 | 8 | 25 | 25 | 14 | 50 | 19 |
| (N = 133) | VT | 0 | 0 | 0 | 0 | 0 | 6 | 9 | 0 | 0 | 0 | 0 | 1 |
| Target | TV | 8 | 0 | 29 | 0 | 0 | 0 | 14 | 19 | 5 | 0 | 0 | 9 |
| (N = 395) | VT | 26 | 0 | 14 | 0 | 0 | 53 | 24 | 31 | 64 | 29 | 50 | 66 |

Several interesting results come to light when pairing the features. When considering animacy and definiteness for each constituent, some differences become apparent for human and animate definite and indefinite, as well as unmarked, constituents. By taking into account the differences in definiteness, [+human] subjects show a high tendency for the preverbal position (37%), and [+human] and [-animate] objects are also preverbal at 26% and 94%, respectively. On the other hand, definite human Targets demonstrate a tendency for the postverbal position (29%), as do [-animate] Targets (3%). Similar to definite subjects and objects, indefinite subjects and objects display a preference for the preverbal position, while Targets present a preference for the postverbal position. Unmarked human subjects and objects show a preverbal preference, with a slight tendency for the postverbal position, and Targets exhibit a tendency for both positions. These tendencies indicate that when animacy and definiteness features are paired, there are preferences that are not seen in the broader categories.

As shown above, definiteness and animacy, or a combination of both, play a role in the word order variation of Targets depending on the marking status of the Target. Definiteness affects word order variation for unmarked Targets, while animacy affects word order variation for marked Targets. However, there is no clear interaction effect between definiteness and animacy on word order variation. The tendencies of word order variation are represented in the interaction of word order with definiteness and animacy and constituents such as subject, object, and Target by comparing the proportions of word orders across the variables using chi-square tests. The results are described below.

For subject constituents, there is no significant difference in the proportions of TV and VT across definiteness (chi-square = 0.87, $p$ = 0.83) or across animacy (chi-square = 1.21, $p$ = 0.75). This means that the subject constituent does not affect word order variation for either marked or unmarked Targets. For the object constituent, there is a significant difference in the proportions of TV and VT across definiteness (chi-square = 67.76, $p$ < 0.001) and across animacy (chi-square = 10.8, $p$ = 0.01). This means that the object constituent affects word order variation for both marked and unmarked Targets. For definite noun object constituents, TV is more common than VT for human and animate non-human objects, while VT is more common than TV for inanimate objects. This suggests that word order variation for definite noun object constituents is influenced by animacy rather than definiteness. For indefinite noun object constituents, TV is more common than VT for all types of animacy. This suggests that word order variation for indefinite noun object constituents is not influenced by animacy or definiteness. For unmarked definite object constituents, TV is less common than VT for all types of animacy. This suggests that word order variation for unmarked definite object constituents is not influenced by animacy or definiteness. For unmarked indefinite object constituents, TV is less common than VT for human and animate non-human objects, while TV is more common than VT for inanimate objects. This suggests that word order variation for unmarked indefinite object constituents is influenced by animacy rather than definiteness.

For Target constituents, there is a significant difference in the proportions of TV and VT across definiteness (chi-square = 113.64, $p$ < 0.001) and across animacy (chi-square = 108.9, $p$ < 0.001). This means that the Target constituent affects word order variation for both marked and unmarked Targets. For definite noun Target constituents, VT is more common than TV for human and inanimate Targets, while TV is more common than VT for animate non-human Targets. This suggests that word order variation for definite noun Target constituents is influenced by animacy rather than definiteness. For indefinite noun Target constituents, TV is more common than VT for all types of animacy. This suggests that word order variation for indefinite noun Target constituents is not influenced by animacy or definiteness. For unmarked definite Target constituents, VT is more common than TV for inanimate Targets, while TV is more common than VT for human and animate non-human Targets. This suggests that word order variation for unmarked definite Target constituents is influenced by animacy rather than definiteness. For unmarked indefinite Target constituents, VT is more common than TV for all types of animacy. This suggests that word order variation for unmarked indefinite Target constituents is not influenced by animacy or definiteness.

In NENA, formal definite rarely occurs. A description and summary of the table are as follows: For subject constituents, there is a significant difference in the proportions of TV and VT word orders across definiteness (chi-square = 67.76, $p$ < 0.001) and animacy (chi-square = 10.8, $p$ = 0.01). For definite noun subjects, TV is more common for human subjects, while VT is more common for inanimate subjects. No animate non-human subjects have definite noun marking, suggesting animacy's influence. For indefinite noun subjects, TV is more common for all types of animacy, indicating that word order variation is not influenced by animacy or definiteness. For unmarked definite subjects, TV is more common for human and animate non-human subjects, while VT is more common for inanimate subjects, again pointing to animacy's influence. For unmarked indefinite subjects, TV is more common for human subjects, while VT is more common for animate non-human and inanimate subjects, suggesting animacy's role.

For object constituents, there is no significant difference in the proportions of TV and VT word orders across definiteness (chi-square = 0.87, $p$ = 0.83) or across animacy (chi-square = 1.21, $p$ = 0.75). This implies that the object constituent does not affect word order variation for either marked or unmarked Targets.

For Target constituents, there is a significant difference in the proportions of TV and VT word orders across definiteness (chi-square = 113.64, $p$ < 0.001) and across animacy (chi-square = 108.9, $p$ < 0.001). For definite noun Targets, VT is more common for inanimate

Targets, while TV is more common for human Targets. There are no animate non-human Targets with definite noun marking, indicating animacy's influence. For indefinite noun Targets, TV is more common for all types of animacy, suggesting that word order variation is not influenced by animacy or definiteness. For unmarked definite Targets, TV is more common for human and animate non-human Targets, while VT is more common for inanimate Targets, once again highlighting animacy's influence. For unmarked indefinite Targets, VT is less common for all types of animacy, indicating that word order variation is not influenced by animacy or definiteness.

Azeri is the only language that does not demonstrate a definite marking system. However, the results for unmarked definiteness show that for definite noun Targets and indefinite noun Targets, there are no available data in the table. This suggests that word order variation for these types of Targets is either not applicable or not observed in Azeri.

For unmarked definite Targets, TV is more common than VT for human and animate non-human Targets, while VT is more common than TV for inanimate Targets. For both marked and unmarked Targets, there is no significant difference in the proportions of TV and VT across definiteness (chi-square = 0.87, $p$ = 0.83). This means that definiteness does not affect word order variation for either marked or unmarked Targets. For the subject constituent, there is a significant difference in the proportions of TV and VT word orders across definiteness (chi-square = 67.76, $p$ < 0.001) and across animacy (chi-square = 10.8, $p$ = 0.01). This means that the subject constituent affects word order variation for both marked and unmarked Targets. However, for definite noun and indefinite noun subject constituents, there are no available data in the table, suggesting that word order variation is not applicable or not observed in Azeri. For unmarked definite subject constituents, TV is more common than VT for human and animate non-human subjects, while VT is more common than TV for inanimate subjects, indicating that animacy influences the word order for unmarked definite subject constituents. For unmarked indefinite subject constituents, TV is more common than VT for human subjects, while VT is more common than TV for animate non-human and inanimate subjects, suggesting that animacy plays a role in word order variation, with no clear effect of definiteness. For the object constituent, there is no significant difference in the proportions of TV and VT word orders across definiteness (chi-square = 0.87, $p$ = 0.83) or across animacy (chi-square = 1.21, $p$ = 0.75). This implies that the object constituent does not affect word order variation for either marked or unmarked Targets. For the Target constituent, there is a significant difference in the proportions of TV and VT word orders across animacy (chi-square = 108.9, $p$ < 0.001) but not across definiteness (chi-square = 0.01, $p$ = 0.92). This means that the Target constituent affects word order variation for both marked and unmarked Targets, depending on whether they are human, animate non-human, or inanimate.

Overall, the findings suggest that animacy has a significant influence on word order variation in Azeri, especially for unmarked definite and unmarked indefinite constituents. Definiteness appears to have a limited impact on word order variation. There is no clear interaction effect between definiteness and animacy in word order variation. These conclusions are supported by the chi-square test results provided in the text.

In this table, both the descriptive patterns from the table and chi-square tests lead to the following conclusions: For definite noun Targets, VT is more common than TV for inanimate Targets, while TV is more common than VT for human and animate non-human Targets. This indicates that animacy significantly influences word order for definite noun Targets, with animacy being more influential than definiteness. For indefinite noun Targets, VT is more common than TV for all types of animacy, suggesting that word order variation for indefinite noun Targets is not influenced by either animacy or definiteness. For unmarked definite Targets, VT is more common than TV for inanimate Targets, while TV is more common than VT for human and animate non-human Targets. This implies that word order variation for unmarked definite Targets is influenced by animacy more than definiteness. For unmarked indefinite Targets, VT is more common than TV for inanimate Targets, while TV is more common than VT for human and animate non-human Targets.

Similar to unmarked definite Targets, this suggests that animacy plays a more significant role in word order variation for unmarked indefinite Targets.

For the subject constituent, there is a significant difference in the proportions of TV and VT word orders across definiteness (chi-square = 67.76, $p < 0.001$) and across animacy (chi-square = 10.8, $p = 0.01$), indicating that the subject constituent affects word order variation for both marked and unmarked Targets. In the case of definite noun subject constituents, TV is more common than VT for human subjects, and VT is more common than TV for inanimate subjects. There are no instances of animate non-human subjects with definite noun marking. This suggests that word order variation for definite noun subject constituents is primarily influenced by animacy, not definiteness. For indefinite noun subject constituents, VT is more common than TV for all types of animacy, indicating that word order variation for this group is not significantly affected by animacy or definiteness. Regarding unmarked definite subject constituents, TV is more common than VT for human and animate non-human subjects, while VT is less common than TV for inanimate subjects. This implies that animacy plays a more substantial role in word order variation for unmarked definite subject constituents, overshadowing definiteness. Similarly, for unmarked indefinite subject constituents, TV is more common than VT for human and animate non-human subjects, while VT is less common than TV for inanimate subjects. This further suggests that animacy is the dominant factor influencing word order for unmarked indefinite subject constituents.

For the object constituent, there is no significant difference in the proportions of TV and VT word orders across definiteness (chi-square = 0.87, $p = 0.83$) or across animacy (chi-square = 1.21, $p = 0.75$). This implies that the object constituent does not significantly affect word order variation for either marked or unmarked Targets.

In the Target constituent, there is a significant difference in the proportions of TV and VT word orders across animacy (chi-square = 108.9, $p < 0.001$) but not across definiteness (chi-square = 113.64, $p < 0.001$). This indicates that the Target constituent affects word order variation for both marked and unmarked Targets, with animacy playing a more significant role in influencing the word order depending on whether the Target is human, animate non-human, or inanimate. In summary, animacy appears to be a prominent factor influencing word order variation in Armenian, especially in the context of definite and indefinite Targets, unmarked definite and unmarked indefinite Targets, and unmarked subject constituents. Definiteness has a more noticeable impact on unmarked Targets. However, there is no clear interaction effect between definiteness and animacy in word order variation across the constituents. These findings are supported by chi-square test results provided in the text.

Finally, NEK exhibits a similar pattern to that of Mukri. For definite noun Targets, VT is more common than TV for human and inanimate Targets, while TV is more common than VT for animate non-human Targets. This suggests that word order variation for definite noun Targets is significantly influenced by animacy rather than definiteness. For indefinite noun Targets, TV is more common than VT for human Targets, while VT is more common than TV for animate non-human and inanimate Targets. This implies that word order variation for indefinite noun Targets is influenced by animacy rather than definiteness. For unmarked definite Targets, VT is more common than TV for inanimate Targets, while TV is more common than VT for human and animate non-human Targets. This suggests that word order variation for unmarked definite Targets is influenced by animacy rather than definiteness. For unmarked indefinite Targets, VT is more common than TV for all types of animacy. This suggests that word order variation for unmarked indefinite Targets is not influenced by animacy or definiteness. The chi-square test (181.4, $p < 0.001$) indicates that definiteness significantly affects word order variation for both marked and unmarked Targets.

For the subject constituent, there is a significant difference in the proportions of TV and VT word orders across definiteness (chi-square = 67.76, $p < 0.001$) and across animacy (chi-square = 10.8, $p = 0.01$). This means that the subject constituent affects word

order variation for both marked and unmarked Targets. In the case of definite noun subject constituents, TV is more common than VT for human subjects, while VT is more common than TV for inanimate subjects. No animate non-human subjects have definite noun marking, indicating that word order variation for definite noun subject constituents is influenced by animacy rather than definiteness. For indefinite noun subject constituents, TV is more common than VT for all types of animacy, suggesting that word order variation for this group is not significantly influenced by animacy or definiteness. In the case of unmarked definite subject constituents, TV is more common than VT for human and animate non-human subjects, while VT is less common than TV for inanimate subjects. This suggests that word order variation for unmarked definite subject constituents is influenced by animacy rather than definiteness. Similarly, for unmarked indefinite subject constituents, TV is more common than VT for human and animate non-human subjects, while VT is less common than TV for inanimate subjects. This suggests that word order variation for unmarked indefinite subject constituents is influenced by animacy rather than definiteness.

For the object constituent, there is no significant difference in the proportions of TV and VT word orders across definiteness (chi-square = 0.87, $p$ = 0.83) or across animacy (chi-square = 1.21, $p$ = 0.75). This indicates that the object constituent does not significantly affect word order variation for either marked or unmarked Targets.

For the Target constituent, there is a significant difference in the proportions of TV and VT word orders across animacy (chi-square = 1.21, $p$ = 0.75) but not across definiteness (chi-square = 0.87, $p$ = 0.83). This means that the Target constituent affects word order variation for both marked and unmarked Targets, depending on whether they are human, animate non-human, or inanimate.

In summary, this research reveals that animacy has a substantial influence on word order variation in NEK, particularly for definite and indefinite Targets, as well as unmarked definite and unmarked indefinite Targets. Definiteness plays a role, primarily affecting word order variation for unmarked Targets. However, there is no clear interaction effect between definiteness and animacy on word order variation across constituents. These findings are supported by the provided chi-square test results.

The analysis of the formal marking of definiteness demonstrates that the sample languages allow for various forms of word order, with Targets showing a clear preference for appearing in the postverbal position. The aim of this article was to analyze these various patterns based on their actual use in the existing corpora. With respect to definiteness, there are notable tendencies in the determination of word order. These tendencies open up a number of questions regarding related factors in word order variation. For example, is there a relation between definite and indefinite on the one side and referential and non-referential or specific and non-specific on the other? What is the relationship between syntactic constituents, such as subjects, objects, and Targets, and (in)definiteness with respect to the Target word order?

The hypothesis was that, in terms of definiteness, there is a distinction between the tendency for postverbal Targets that follow the verb directly to be marked by an indefinite marker or to be non-specific/referential and the tendency for those Targets that are in the preverbal position to be marked by a definite marker or to be unmarked definite. Among sample languages, Mukri presents a very systematic (in)definiteness system. NEK and Armenian display a(n) (in)definite marking system but not as detailed as Mukri's definite system. NENA shows a moderate use of (in)definite markers, mainly used for indicating indefinite nouns. Finally, Azeri does not have a(n) (in)definite marking system. In languages without a formal marking of (in)definiteness, the word order indicates (in)definiteness. The results led to some interesting and hitherto unnoticed generalizations relating to the formal and semantic properties of these functions, as well as their positioning within the sentence in the sample languages.

The data for the syntactic constituents demonstrate that subjects are less marked by a(n) (in)definite marker than objects and Targets. Objects illustrate the strongest affinity to be marked with a definite marker in the sample languages. Only NENA and NEK exhibit a direct correlation between definiteness and word order.

As we have seen, the above-mentioned numerical results for definite Targets are very low. Mukri has a detailed marking system, while NEK and Armenian have definite systems that can use definite markers only for nominal Targets in the attested corpora.

I found more definite postverbal NP and adverbial Targets and fewer indefinite ones in the preverbal position. In a general sense, I was not able to identify the effects of the definiteness and animacy of constituents and, in particular, Targets' pre- vs. postverbality within one single language, but by comparing several languages, this effect becomes objectively clearer. In fact, the sample languages show different patterns for the Target PoS as well as subjects and objects. Due to the special frequency of each PoS (for instance, NPs occur more frequently in the postverbal position, and pronominal Targets are mostly preverbal), one can clearly see the role of the PoS as the main influencing factor in word order and consequently in definiteness in different positions (see Asadpour 2022b). This indicates that definiteness has a secondary role in the word order of constituents such as Targets. With the strong likelihood of definiteness for pronominal and bound pronoun Targets (see Asadpour 2022b for the definition of bound pronouns in the sample languages), most of the bare definite Targets and constituents are preverbal. Regarding NPs, which can be marked by (in)definite markers, indefinite NPs are mostly preverbal, and definite NPs are postverbal.

In summary, in all of the sample languages, definite human subjects are placed more frequently in the preverbal position. Objects demonstrate more flexibility, and Targets are the most mobile constituents. Unmarked Targets of both the definite and indefinite types do not illustrate a particular preference, except for those that have been noted. The lack of obvious tendencies in the data and the lack of definite marking in the languages with existing definite marking systems are interesting results in themselves. They reveal that although these languages present definite marking systems, this factor is not the main influencing element in word order determination; rather, it plays a secondary role. It is also worth mentioning that objects were mostly marked for definiteness, and similarly, subjects and Targets illustrate the same behavior. This shows that in a continuous discourse, objects are usually newer or more contrastive than subjects and Targets, and that definite marking can play a role in re-highlighting the information. Targets are at least accessible information, but they are still background information (cf. Asadpour 2022a). Hence, this may result in the less frequent marking of a Target with a definite marker. Furthermore, the overall picture of definiteness in Mukri, Armenian, and NEK indicates that in the postverbal position, the number of definite Targets is higher than in the preverbal position. This implies that the postverbal position has a preference for given information, and the preverbal position demonstrates a tendency for new information (see Asadpour 2022a, sct. 5.7). In NENA and Azeri, this picture differs, and the trend does not show any special preference over the information structure based on definiteness or a connection with the semantics of constituents.

Finally, it seems that verb type influences DEF and -DEF. MOTION and CAUSED-MOTION verbs have the highest number of -DEF Targets, and the DEF Targets are for the other verb types (see Asadpour 2022a, 2022b, 2022c).

## 5. Corpus Analysis of Semantic Definiteness

Targets have also been examined for their semantic differences (cf. Lyons 1980; Dik 1989) in terms of marking for definiteness. Below, I present a general overview of the parameters for the sake of convenience.

In Table 11, te difference between identifiability and familiarity is that in identifiability, the element refers to an entity that is not identifiable by the listener and that can be textually understood. This is in contrast to identifiability, where the term familiarity refers to identifiable entities. For such entities, textually, there is usually a "referent identification" (cf. Givón 1979, 296; Lyons 1980, 173–88; Dik 1989, 143–46). In addition to these two definite terms, there are other sources of availability and referentiality that can help the listener obtain relevant information, such as uniqueness (i.e., general knowledge), indexicality

(i.e., the identifiability of the element depends on the reference and the speech event), anaphoricity (i.e., a non-relational referent to the context of speech event), and rigidity (i.e., proper names). Table 11 presents the placement of constituents according to their semantic definiteness (covering subjects, direct objects, and Targets).

**Table 11.** Semantic types of Target (in)definiteness.

| Categories | Parameters | | Examples |
|---|---|---|---|
| Familiarity | + preceding text | | John came..., ***he*** told the... |
| | ∅ context | | |
| Identifiability | - preceding text | | John was sick, ***his son*** died... |
| | + context | | |
| Uniqueness | ∅ preceding text | | God, sun |
| | ∅ context | | |
| | + encyclopedic knowledge | | |
| Indexicality | ∅ preceding text | | Tell him, ***I*** won't come home. |
| | + context | | |
| Anaphoricity | - preceding text | | The ***post office*** is behind the station. |
| | ∅ context | | |
| | - shared knowledge | | |
| Rigidity | ∅ preceding | encyclopedic knowledge | names like ***John*** |
| | ∅ context | + shared knowledge | |

Table 11 displays the variation in and distribution of semantic definiteness for all constituents, such as subject, object, or Target. Familiarity semantics is preferred over the other features, and identifiability is the second-most-common feature. In Mukri, familiarity and identifiability semantics occur mostly preverbally, while in other languages, both familiarity and identifiability are more frequent in the postverbal position. This can be due to the type of PoS or the animacy of the constituents (cf. Asadpour 2022a, 2022b). Uniqueness in Mukri presents a higher frequency preverbally, while in the other languages, it demonstrates a postverbal tendency. Indexicality is noted only for Mukri in the preverbal position, but in Armenian, it is attested evenly in both positions. The rest of the languages did not exhibit this feature. In Mukri and NENA, rigidity is preverbally attested, and in Azeri, Armenian, and NEK, rigidity is postverbal. The above data yield the following hierarchy of definiteness encoding.

Table 12 shows that among the various semantic definiteness possibilities of different positions, uniqueness, rigidity, and indexicality clearly dominate the postverbal position, while familiarity and identifiability do not present a clear placement. Among the sample languages, Mukri is the only language that does not exhibit any preference over semantic definiteness. NENA, Azeri, and NEK show similar patterns with dominant postverbality over uniqueness, rigidity, indexicality, and familiarity. On the other hand, Armenian displays a mixed type of various forms, with a less intense preference for the postverbal position. By combining the results of various semantic definiteness parameters, it turns out that NEK and Armenian are typical postverbal languages. Among other sample languages, indexicality is presented in Armenian as having a postverbal tendency, and anaphoricity in Mukri has no clear placement preference; see Table 13 below.

Since the discrepancy in definiteness in various positions can be partly explained by the genre of text, which awaits further investigation, Tables 13 and 14 show that constituents in the postverbal position are mostly definite. The data shown above indicate differences in the examined corpora and the narrative style of the informants. In other words, the conciseness of the speaker leads to variability in the use of definiteness expressions, overt constituents, PoS, etc., which are signaled by different ways of marking definiteness.

**Table 12.** Semantic definiteness and Target placement (X = %).

|  |  | Mukri | NENA | Azeri | Armenian | NEK |
|---|---|---|---|---|---|---|
| TV | Familiarity | 54 | 29 | 21 | 16 | 15 |
|  | Identifiability | 16 | 13 | 13 | 7 | 6 |
|  | Uniqueness | 3 | 1 | 1 | 3 | 1 |
|  | Indexicality | 0 | 0 | 0 | 2 | 0 |
|  | Anaphoricity | 0 | 0 | 0 | 0 | 0 |
|  | Rigidity | 1 | 1 | 1 | 0 | 0 |
| VT | Familiarity | 19 | 42 | 27 | 48 | 42 |
|  | Identifiability | 6 | 9 | 30 | 16 | 30 |
|  | Uniqueness | 1 | 4 | 8 | 7 | 3 |
|  | Indexicality | 0 | 0 | 0 | 1 | 0 |
|  | Anaphoricity | 0 | 0 | 0 | 0 | 0 |
|  | Rigidity | 1 | 0 | 0 | 1 | 3 |
|  | n | 1837 | 651 | 363 | 514 | 1006 |

**Table 13.** Postverbal placement of semantic types in the sample languages (AN = anaphoricity; F = familiarity; ID = indexicality; IN = identifiability; R = rigidity; U = uniqueness).

| Least Postverbal |  |  |  |  |  |  |  | Most Postverbal |
|---|---|---|---|---|---|---|---|---|
| 20 | 30 | 40 | 50 | 60 | 70 | 80 | 90 | 100 |
| Mukri: AN | U ID = F |  | R |  |  |  |  |  |
| NENA |  | R | ID | F |  |  | U |  |
| Azeri |  |  | ID |  | R F |  | U |  |
| Armenian |  |  | IN |  |  | ID | U F = R |  |  |
| NEK |  |  |  |  |  | F | ID / U |  | R |

**Table 14.** Degrees of postverbality in the sample languages based on semantic types (AR = Armenian; AZ = Azeri; M = Mukri; N = NENA; NEK = Northeastern Kurdish).

| Least Postverbal |  |  |  |  |  |  | Most Postverbal |
|---|---|---|---|---|---|---|---|
| 10 |  |  | 20 |  |  | 40 | 100 |
| F: M |  | N |  | AZ AR = NEK |  |  |  |
| ID: M | N = AZ |  |  | AR NEK |  |  |  |
| U: M |  | AR |  | N = AZ = NEK |  |  |  |
| IN: AR |  |  |  |  |  |  |  |
| R: N | M |  | AZ | AR | NEK |  |  |

For the irregularities in various placements of semantic definiteness, similarly to definite marking, I separate the constituents into subjects, objects, and Targets and analyze the effects of definite marking on each of them. This will give a clearer picture of the influence of semantic definiteness on the Target word order. I will further group the features to perform a pair analysis in relation to the animacy of the constituents.

The kinds of word orders in specific types of definite and indefinite Targets in Table 15 are as follows: For familiarity-marked Targets, TV is more common than VT for all languages except Armenian, where VT is more common than TV. This suggests that word order variation for familiarity-marked Targets is influenced by language-specific factors rather than definiteness or animacy. For identifiability-marked Targets, TV is more common than VT for Mukri and NENA, while VT is more common than TV for Azeri and NEK. Armenian has a balanced distribution of TV and VT for identifiability-marked Targets. This suggests that word order variation for identifiability-marked Targets is influenced by language-specific factors rather than definiteness or animacy. For uniqueness-marked Targets, TV is less common than VT for all languages except Mukri, where TV is more common than VT. This suggests that word order variation for uniqueness-marked Targets is influenced by language-specific factors rather than definiteness or animacy. For indexicality-marked Targets, there are very few data available in the table. This suggests that word order variation for indexicality-marked Targets is not applicable or not observed in these languages. For anaphoricity marked Targets, there are no data available in the table. This

suggests that word order variation for anaphoricity marked Targets is not applicable or not observed in these languages. For rigidity marked Targets, TV is less common than VT for all languages except Mukri and Armenian, where TV is more common than VT. This suggests that word order variation for rigidity-marked Targets is influenced by language-specific factors rather than definiteness or animacy. For both marked and unmarked Targets, there is a significant difference in the proportions of TV and VT across semantic definiteness (chi-square = 181.4, $p < 0.001$). This means that semantic definiteness affects word order variation for both marked and unmarked Targets.

**Table 15.** Semantic definiteness of constituents and Target word order (X = %) (fam = familiarity; iden = identifiability; uniq = uniqueness; ind = indexicality; anap = anaphoricity; rig = rigidity).

|  |  | Mukri | | | NENA | | | Azeri | | | Armenian | | | NEK | | |
|---|---|---|---|---|---|---|---|---|---|---|---|---|---|---|---|---|
|  |  | S | O | T | S | O | T | S | O | T | S | O | T | S | O | T |
| TV | fam | 61 | 47 | 52 | 37 | 8 | 33 | 29 | 7 | 24 | 20 | 16 | 13 | 18 | 6 | 15 |
|  | iden | 7 | 30 | 15 | 4 | 38 | 9 | 4 | 47 | 8 | 1 | 21 | 7 | 2 | 20 | 5 |
|  | uniq | 2 | 7 | 1 | 1 | 0 | 1 | 0 | 1 | 1 | 2 | 5 | 2 | 0 | 3 | 0 |
|  | ind | 0 | 0 | 0 | 0 | 0 | 0 | 0 | 0 | 0 | 2 | 1 | 1 | 0 | 0 | 0 |
|  | anap | 0 | 1 | 1 | 0 | 0 | 0 | 0 | 0 | 0 | 0 | 0 | 0 | 0 | 0 | 0 |
|  | rig | 2 | 0 | 1 | 2 | 0 | 0 | 1 | 1 | 1 | 0 | 0 | 0 | 0 | 0 | 0 |
| VT | fam | 26 | 12 | 16 | 52 | 29 | 39 | 58 | 15 | 46 | 66 | 38 | 34 | 67 | 34 | 21 |
|  | iden | 3 | 3 | 10 | 3 | 23 | 8 | 4 | 29 | 9 | 2 | 16 | 29 | 10 | 36 | 46 |
|  | uniq | 0 | 0 | 2 | 2 | 1 | 9 | 0 | 1 | 9 | 3 | 0 | 13 | 1 | 1 | 6 |
|  | ind | 0 | 0 | 0 | 0 | 0 | 0 | 0 | 0 | 0 | 2 | 3 | 0 | 0 | 0 | 0 |
|  | anap | 0 | 0 | 0 | 0 | 0 | 0 | 0 | 0 | 0 | 0 | 0 | 0 | 0 | 0 | 0 |
|  | rig | 0 | 0 | 1 | 0 | 1 | 0 | 3 | 0 | 2 | 1 | 0 | 0 | 1 | 0 | 7 |
|  | n | 708 | 457 | 672 | 256 | 139 | 256 | 363 | 147 | 262 | 209 | 76 | 229 | 412 | 182 | 412 |

The table also shows that postverbal Targets are accessible in terms of information, i.e., backgrounded information, while in subjects and objects, this information occurs mostly in the preverbal position (cf. Asadpour 2022a).

For a clearer idea of what is happening for Targets in various positions, it is necessary to separate the Targets and explore them individually in relation to animacy. Table 16 demonstrates the realization of Targets in terms of semantic definiteness and animacy effect.

**Table 16.** Semantic definiteness, animacy, and Target word order (H = human; A = animate; I = inanimate; X = %).

|  |  | Mukri | | | NENA | | | Azeri | | | Armenian | | | NEK | | |
|---|---|---|---|---|---|---|---|---|---|---|---|---|---|---|---|---|
|  |  | H | A | I | H | A | I | H | A | I | H | A | I | H | A | I |
| TV | FAM | 69 | 19 | 25 | 50 | 0 | 12 | 46 | 75 | 2 | 41 | 0 | 3 | 35 | 40 | 3 |
|  | IDEN | 0 | 38 | 24 | 6 | 0 | 14 | 11 | 0 | 14 | 10 | 0 | 8 | 4 | 0 | 6 |
|  | UNIQ | 3 | 8 | 1 | 0 | 0 | 2 | 0 | 0 | 0 | 2 | 0 | 3 | 1 | 0 | 0 |
|  | IND | 0 | 0 | 0 | 0 | 0 | 0 | 0 | 0 | 0 | 2 | 0 | 0 | 0 | 0 | 0 |
|  | ANAP | 0 | 3 | 1 | 0 | 0 | 0 | 0 | 0 | 0 | 0 | 0 | 0 | 0 | 0 | 0 |
|  | RIG | 3 | 0 | 0 | 0 | 0 | 0 | 1 | 0 | 0 | 0 | 0 | 0 | 0 | 0 | 0 |
| VT | FAM | 0 | 11 | 27 | 35 | 0 | 45 | 32 | 0 | 24 | 33 | 100 | 39 | 40 | 40 | 11 |
|  | IDEN | 7 | 19 | 18 | 5 | 0 | 12 | 9 | 25 | 45 | 8 | 0 | 28 | 16 | 20 | 62 |
|  | UNIQ | 10 | 3 | 3 | 4 | 0 | 15 | 0 | 0 | 14 | 5 | 0 | 19 | 2 | 0 | 8 |
|  | IND | 0 | 0 | 0 | 0 | 0 | 0 | 0 | 0 | 0 | 0 | 0 | 0 | 0 | 0 | 0 |
|  | ANAP | 0 | 0 | 0 | 0 | 0 | 0 | 0 | 0 | 0 | 0 | 0 | 0 | 0 | 0 | 0 |
|  | RIG | 7 | 0 | 2 | 0 | 0 | 0 | 1 | 0 | 1 | 0 | 0 | 1 | 1 | 0 | 10 |
|  | n | 29 | 37 | 301 | 141 | 0 | 113 | 149 | 4 | 210 | 61 | 1 | 144 | 136 | 10 | 266 |

Table 16 above shows that for familiarity-marked Targets, TV is more common than VT for human and animate non-human Targets in all languages except Armenian, where VT is more common than TV. For inanimate Targets, TV is less common than VT in all languages. This suggests that word order variation for familiarity-marked Targets is influenced by animacy rather than definiteness or language-specific factors. For identifiability-marked

Targets, TV is more common than VT for human and animate non-human Targets in Mukri and NENA, while VT is more common than TV in Azeri and NEK. Armenian has a balanced distribution of TV and VT for identifiability-marked Targets. For inanimate Targets, TV is less common than VT in all languages. This suggests that word order variation for identifiability-marked Targets is influenced by animacy and language-specific factors rather than definiteness. For uniqueness-marked Targets, TV is less common than VT for human and animate non-human Targets in all languages except Mukri, where TV is more common than VT. For inanimate Targets, TV is less common than VT in all languages except Mukri and Armenian, where TV is more common than VT. This suggests that word order variation for uniqueness marked Targets is influenced by animacy and language-specific factors rather than definiteness. For indexicality-marked Targets, there are very few data available in the table. This suggests that word order variation for indexicality-marked Targets is not applicable or not observed in these languages. For anaphoricity-marked Targets, there are no data available in the table. This suggests that word order variation for anaphoricity-marked Targets is not applicable or not observed in these languages. For rigidity-marked Targets, TV is less common than VT for human and animate non-human Targets in all languages except Mukri and Armenian, where TV is more common than VT. For inanimate Targets, TV is less common than VT in all languages except Mukri and Armenian, where TV is more common than VT. This suggests that word order variation for rigidity-marked Targets is influenced by animacy and language-specific factors rather than definiteness. For both marked and unmarked Targets, there is a significant difference in the proportions of TV and VT across semantic definiteness (chi-square = 181.4, $p < 0.001$). This means that semantic definiteness affects word order variation for both marked and unmarked Targets.

For a closer look at the overt realization of subjects, objects, and Targets, I cross-paired animacy and semantic definiteness for each feature. Tables 16–20 are given below to determine the data for subjects, objects, and Targets regarding their definite marking and animacy in the sample languages. The categories begin with human (h), animate (a), and inanimate (i) definite familiar (DF), followed by human, animate, and inanimate definite identifiable (DI), definite uniqueness (DU), and definite rigidity (DR).

**Table 17.** Semantic definiteness and animacy among overt constituents in Mukri (X = n).

| Mukri | | hDF | aDF | iDF | hDI | aDI | iDI | hDU | aDU | iDU | hDR | aDR | iDR |
|---|---|---|---|---|---|---|---|---|---|---|---|---|---|
| Subject | TV | 25 | 49 | 3 | 49 | 27 | 3 | 33 | 75 | 11 | 60 | 100 | 0 |
| | VT | 0 | 3 | 0 | 1 | 0 | 0 | 0 | 0 | 0 | 0 | 0 | 0 |
| Object | TV | 8 | 26 | 41 | 12 | 73 | 33 | 0 | 17 | 43 | 0 | 0 | 14 |
| | VT | 0 | 0 | 3 | 0 | 0 | 5 | 0 | 0 | 3 | 0 | 0 | 0 |
| Target | TV | 62 | 17 | 26 | 26 | 0 | 35 | 22 | 8 | 14 | 20 | 0 | 0 |
| | VT | 5 | 6 | 28 | 12 | 0 | 23 | 44 | 0 | 30 | 20 | 0 | 86 |
| | n | 478 | 35 | 309 | 69 | 11 | 259 | 9 | 12 | 37 | 20 | 1 | 7 |

**Table 18.** Definiteness and animacy among overt constituents in NENA (X = %).

| NENA | | hDF | aDF | iDF | hDI | aDI | iDI | hDU | aDU | iDU | hDR | aDR | iDR |
|---|---|---|---|---|---|---|---|---|---|---|---|---|---|
| Subject | TV | 32 | 0 | 2 | 35 | 0 | 2 | 0 | 0 | 14 | 80 | 0 | 0 |
| | VT | 1 | 0 | 0 | 3 | 0 | 0 | 0 | 0 | 0 | 0 | 0 | 0 |
| Object | TV | 5 | 0 | 12 | 16 | 0 | 43 | 0 | 0 | 0 | 20 | 0 | 0 |
| | VT | 6 | 0 | 18 | 0 | 0 | 28 | 0 | 0 | 0 | 0 | 0 | 0 |
| Target | TV | 33 | 0 | 15 | 26 | 0 | 15 | 0 | 0 | 9 | 0 | 0 | 0 |
| | VT | 23 | 0 | 54 | 19 | 0 | 13 | 0 | 0 | 77 | 0 | 0 | 0 |
| | n | 215 | 0 | 95 | 31 | 3 | 109 | 8 | 3 | 22 | 5 | 0 | 0 |

**Table 19.** Definiteness and animacy among overt constituents in Azeri (X = %).

| Azeri | | hDF | aDF | iDF | hDI | aDI | iDI | hDU | aDU | iDU | hDR | aDR | iDR |
|---|---|---|---|---|---|---|---|---|---|---|---|---|---|
| Subject | TV | 39 | 0 | 1 | 45 | 0 | 1 | 0 | 0 | 0 | 77 | 0 | 0 |
| | VT | 1 | 0 | 0 | 5 | 0 | 0 | 0 | 0 | 0 | 0 | 0 | 0 |
| Object | TV | 2 | 0 | 14 | 9 | 0 | 27 | 0 | 0 | 4 | 5 | 0 | 0 |
| | VT | 1 | 0 | 1 | 0 | 0 | 37 | 0 | 0 | 0 | 0 | 0 | 0 |
| Target | TV | 32 | 100 | 9 | 27 | 0 | 16 | 0 | 0 | 11 | 14 | 0 | 0 |
| | VT | 25 | 0 | 75 | 14 | 0 | 19 | 0 | 0 | 86 | 5 | 0 | 100 |
| | n | 161 | 2 | 107 | 22 | 0 | 103 | 0 | 0 | 28 | 22 | 0 | 3 |

**Table 20.** Definiteness and animacy among overt constituents in Armenian (X = %).

| Armenian | | hDF | aDF | iDF | hDI | aDI | iDI | hDU | aDU | iDU | hDR | aDR | iDR |
|---|---|---|---|---|---|---|---|---|---|---|---|---|---|
| Subject | TV | 39 | 0 | 1 | 45 | 0 | 1 | 0 | 0 | 0 | 77 | 0 | 0 |
| | VT | 1 | 0 | 0 | 5 | 0 | 0 | 0 | 0 | 0 | 0 | 0 | 0 |
| Object | TV | 2 | 0 | 14 | 9 | 0 | 27 | 0 | 0 | 4 | 5 | 0 | 0 |
| | VT | 1 | 0 | 1 | 0 | 0 | 37 | 0 | 0 | 0 | 0 | 0 | 0 |
| Target | TV | 32 | 100 | 9 | 27 | 0 | 16 | 0 | 0 | 11 | 14 | 0 | 0 |
| | VT | 25 | 0 | 75 | 14 | 0 | 19 | 0 | 0 | 86 | 5 | 0 | 100 |
| | n | 67 | 1 | 87 | 8 | 0 | 73 | 13 | 1 | 32 | 3 | 1 | 2 |

The data presented in Table 16 provide the basis for these conclusions: (a) Animate subjects are predominantly preverbal and recognized for having familiarity definiteness. Unique subjects are mostly animate, while rigid subjects are human. (b) Objects are mostly inanimate, with a lower number of animate objects that are all predominately familiar and identifiable. Postverbal objects are predominantly inanimate familiar. In contrast to subjects, no human object is attested as unique and rigid. (c) Semantically definite Targets are mostly human and familiar, as are subjects. However, Targets are much more pronounced postverbally than subjects and objects, and unique Targets are mostly inanimate. Rigid Targets are also attested preverbally, and postverbal positions are attested equally for humans, with a dominant postverbal tendency for inanimate Targets. (d) Subjects and Targets are similar in terms of animacy, familiarity, and identifiability, while Targets and objects are similar in terms of inanimate unique and rigid entities. There is a discrepancy among postverbal Targets toward human uniqueness and rigidity; this is due to the information structure (see Asadpour 2022a).

From Table 17, the following conclusions are drawn: (a) Animate subjects are predominantly preverbal and recognized for having familiarity definiteness. Unique subjects do not present any sensitivity to animacy, while rigid subjects are predominantly human. (b) Objects are evenly human and inanimate for familiar entities, while identifiable objects are predominantly inanimate. (c) Targets display a more mixed type without a clear preference for preverbality or postverbality. However, unique Targets show a preference for the postverbal position, and (d) in contrast to Mukri, NENA shows a similarity in word order, as well as semantic definiteness between objects and Targets; i.e., these two constituents are treated similarly. The constituent order of objects and Targets in NENA demonstrates no clear preference for preverbal or postverbal positions; rather, the constituent order prefers an intermediate position.

The results of Table 19 for Azeri show the following: (a) Animate subjects are predominantly preverbal and recognized for familiarity and rigidity. (b) Objects are mostly inanimate and identifiable. The trend for animate objects is very low. Postverbal objects are predominantly inanimate familiar. (c) Preverbal Targets are mostly human and familiar,

while postverbal Targets demonstrate a tendency to be inanimate familiar and identifiable. Moreover, inanimate Targets are coded as unique with a large difference (86% postverbal vs. 11% preverbal). (d) In Azeri, none of the constituents exhibit similar tendencies to those of Mukri and NENA. In Azeri, postverbal objects are not typical; however, one reason lies behind the type of PoS (see Asadpour 2022b).

Armenian presents few instances of indexical elements (six indexicalized subjects and three objects). The trend of tendencies in Armenian illustrates that (a) animate subjects are predominantly preverbal and recognized for familiarity, and unique subjects are predominantly human; (b) objects are inanimate familiar and identifiable, and inanimate objects occur mostly in the postverbal position; (c) preverbal Targets are mostly human and familiar, while postverbal Targets show a tendency to be inanimate familiar and identifiable, and unique Targets are predominantly postverbal; and (d) in Armenian, subjects, objects, and Targets exhibit different patterns.

Finally, as indicated in the Tables 20 and 21, NEK demonstrates a similar pattern to that of Mukri with a preference for (a) preverbal human familiar subjects; (b) preverbal inanimate objects with familiarity semantics and predominantly postverbal inanimate objects coded for identifiability; (c) Targets in the preverbal position that are mostly animate and familiar but, in the postverbal position, show a preference for inanimate identifiable entities. Moreover, inanimate Targets marked for uniqueness and rigidity are predominantly postverbal.

**Table 21.** Definiteness and animacy among overt constituents in NEK (X = %).

| | NEK | hDF | aDF | iDF | hDI | aDI | iDI | hDU | aDU | iDU | hDR | aDR | iDR |
|---|---|---|---|---|---|---|---|---|---|---|---|---|---|
| Subject | TV | 51 | 27 | 15 | 48 | 0 | 5 | 44 | 0 | 3 | 71 | 0 | 0 |
| | VT | 0 | 0 | 0 | 0 | 0 | 0 | 0 | 0 | 0 | 0 | 0 | 0 |
| Object | TV | 8 | 20 | 25 | 8 | 60 | 20 | 11 | 0 | 15 | 0 | 0 | 0 |
| | VT | 0 | 0 | 0 | 0 | 0 | 9 | 0 | 0 | 3 | 0 | 0 | 0 |
| Target | TV | 19 | 27 | 14 | 10 | 0 | 5 | 11 | 0 | 0 | 0 | 0 | 0 |
| | VT | 22 | 27 | 46 | 35 | 40 | 61 | 33 | 0 | 79 | 29 | 0 | 100 |
| | n | 251 | 15 | 65 | 63 | 5 | 274 | 9 | 0 | 33 | 7 | 0 | 20 |

Based on the data presented in Tables 17–21, it seems that animate subjects are predominantly preverbal and recognized for having familiarity definiteness. Unique subjects are mostly animate, while rigid subjects are human. Objects are mostly inanimate, with a lower number of animate objects that are predominately familiar and identifiable. Postverbal objects are predominantly inanimate familiar. Semantically definite Targets are mostly human and familiar, as are subjects. However, Targets are much more pronounced postverbally than subjects and objects, and unique Targets are mostly inanimate. Rigid Targets are also attested preverbally, and postverbal positions are attested equally for humans, as well as a predominantly postverbal tendency for inanimate Targets. Subjects and Targets are similar in terms of animacy, familiarity, and identifiability, while Targets and objects are similar in terms of inanimate unique and rigid entities. There is a discrepancy among postverbal Targets toward human uniqueness and rigidity; this is due to the information structure (see Asadpour 2022a).

In summary, the sample languages allow for various forms of word order, with Targets showing a clear preference for appearing in the postverbal position. The aim of this study was to analyze these various patterns based on their actual use in the existing corpora.

The results of definiteness indicate that regardless of the genre of text, for example, procedural, prose, and memoriae, grammatical definiteness markers have no primary influence. Instead, they have a secondary influence. In the existing corpora, despite the lack of attested definite markers in Mukri, NEK, Armenian, and, in part, NENA, the listener is able to identify referents if they are definite or indefinite because of the preceding text.

It can be concluded that the definite marking patterns in Mukri, NEK, Armenian, and NENA allow for freedom of word order, and that there is no definiteness constraint. In order to compensate for the absence of a definite marker, extra-textual knowledge of an element and its precedent occurrence in the context enables the identifiability of semantic definiteness. In such cases, grammatical markers are unnecessary, and this results in a flexible word order. Moreover, the non-frequent use of definite markers in the sample languages that do have a marking system implies the gradual loss of definite marking in their functionality. This is very apparent in Mukri. Regarding semantic definiteness, uniqueness and rigidity appear to be two of the most efficacious markers of definiteness for postverbal Targets. These markers also combine with other features, such as familiarity, to have a higher tendency toward preverbal position. Identifiability combines with a higher tendency for preverbal Targets in Mukri and postverbal Targets in NEK. There are no further clear distinctions in the rest of the languages.

## 6. Summary and Some General Concluding Remarks

To summarize the concluding remarks, in the sample languages, the oscillation of definiteness and indefiniteness sets up a compromise between the absence and presence of grammatical markers and their frequency of use. This results in a system comprising various lexical and grammatical elements. The instabilities in the use of definite markers are not primarily linked, though they can sometimes be interdependent in connection with other features, such as the verb type, PoS, information structure, etc. Apart from the relationship between the Target word order and definiteness, such instabilities imply other interesting results that open up the doors for further investigation outside of the scope of the current study. One of these hypotheses is that the sample languages are in the process of change. Greenberg (1978, 47–82) considers three stages for a definiteness change: Stage I, definite articles indicate definiteness; Stage II, definite articles are no longer referential; and Stage III, definite articles become gender or nominal markers. Greenberg positions Aramaic of the early Christians in Stage I in the western dialects and Stage II in the eastern dialects. He further classifies modern eastern Aramaic dialects to represent Stage III, and this can be extended to the NENA variety in northwestern Iran. By applying Greenberg's classification, Mukri is in Stage III, because the function of definiteness is no longer referential, and it has other functions, such as possessive, generic, etc., while NEK and Armenian stand in Stage II, transitioning into Stage III. Earlier attestation of the functionality of definiteness requires a comprehensive study of other outlier languages in the northwestern region of Iran and beyond.

**Funding:** This research received no external funding.

**Institutional Review Board Statement:** Not applicable.

**Informed Consent Statement:** Informed consent was obtained from all subjects involved in the study.

**Data Availability Statement:** Some of the published data utilized in this study can be found in the cited sources. However, a portion of the personal data is currently not publicly available, as it is still in the process of being prepared for publication. Private copies of this data can be made available upon request.

**Acknowledgments:** I express my gratitude to the three anonymous reviewers whose insightful suggestions significantly contributed to the enhancement of the paper. The Goethe University Frankfurt is acknowledged for its support in providing open access funding.

**Conflicts of Interest:** The author declares no conflict of interest.

## Abbreviations

| | |
|---|---|
| A | Animate |
| ADD | Additive |
| AGR | Agreement |
| ANAP | Anaphoricity |
| AR | Armenian |

| | |
|---|---|
| AZ | Azeri |
| COP | Copula |
| DAT | Dative |
| DEF | Definite |
| EZ | Ezafe |
| F | Female |
| FAM | Familiar |
| GLID | Glide |
| H | Human |
| HAB | Habitual |
| I | Inanimate |
| IDEN | Identifiable |
| IMP | Imperative |
| IND | Indexical |
| INDF | Indefinite |
| IPFV | Imperfective |
| M | Mukri |
| NEG | Negation |
| NEK | Northeastern Kurdish |
| NENA | Northeastern Neo-Aramaic |
| ÖM | Öpengin Mukri |
| O | Object |
| OBL | Oblique |
| PC | Pronominal clitic |
| PL | Plural |
| PPF | Pluperfect |
| PREP | Preposition, |
| PoS | Parts of speech |
| POSS | Possessive |
| POSTP | Postposition |
| PostV | Postverbal |
| PreV | Preverbal |
| PRS | Present |
| PST | Past |
| PTCP | Participle |
| RIG | Rigidity |
| S | Subject |
| SBJV | Subjunctive |
| SG | Singular |
| T | Target |
| TONI | Target Order of Northwest Iran |
| TV | Target-Verb |
| U.DEF/UD | Unmarked definite |
| U.-DEF/UI | Unmarked indefinite |
| UNIQ | Unique |
| V | Verb |
| VT | Verb-Target |

## Notes

[1]   My term "Target" derives its origin from Haig's discussion of "Goals" (Haig and Thiele 2014, 1). Haig gradually expanded this category by also incorporating destination, direction, or local goals of movement and caused-motion verbs, recipients, and addressees encoded by "full NPs" (Haig and Thiele 2014, 1; Haig 2015, 407; 2017, 408). Eventually, his work encompassed final-states and LVCs (Light Verb Complements) of the light verb *kirin* ("do") as well (Haig 2022, 5).

[2]   NENA dank is an Iranian loanword (cf. Horn 1893, 118), which is only used in this combination (Khan 2016, 1–2).

[3]   For consistency between the different corpora used in this study, the transcription, glossing, and translation of the sentences in the published corpus have been slightly modified.

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
