# Peer review of "A Corpus Analysis of the Effects of Definiteness and Animacy on Word Order Variation"

_languages, doi:10.3390/languages8040279_

Round 1

Reviewer 1 Report

Comments and Suggestions for Authors

The article is well-written, clear as far as style is concerned. It is also theoretically sound. The statistics and data are well presented ans easy to follow. It also provides an insightful contribution to syntactic typology and definiteness related concepts. Three micro suggestions are included in the pdf of the article

Author Response

Dear Reviewer,

I want to extend my sincere appreciation for your constructive feedback, which has been invaluable in identifying and addressing certain issues within the manuscript. I have taken your suggestions to heart and made the necessary revisions accordingly.

To facilitate your review, I have also included the revised version of the manuscript, along with a PDF document that contains my responses to your comments, directly appended beneath each comment.

Your input has greatly contributed to the improvement of the paper, and I am grateful for your time and effort in evaluating my work. Please feel free to reach out if you have any further questions or require additional clarifications.

Thank you once again for your valuable insights.

Warm regards,

Reviewer 2 Report

Comments and Suggestions for Authors

Comments for minor revision are found in attached file; especially data set could be described in more detail (size, date etc).

Author Response

Dear Reviewer,

I want to extend my sincere appreciation for your constructive feedback, which has been invaluable in identifying and addressing certain issues within the manuscript. I have taken your suggestions to heart and made the necessary revisions accordingly.

To facilitate your review, I have also included the revised version of the manuscript, along with a PDF document that contains my responses to your comments, directly appended beneath each comment.

Your input has greatly contributed to the improvement of the paper, and I am grateful for your time and effort in evaluating my work. Please feel free to reach out if you have any further questions or require additional clarifications.

I added a one document which includes your previous comments with my responses and the edited version of the article.

Thank you once again for your valuable insights.

Warm regards,

Reviewer 3 Report

Comments and Suggestions for Authors

This paper prsents a corpus view of word order across several languages spoken in Iran and its neighboring countries with a certain level of contact between them. The paper investigates the correlation between definiteness, animacy and word order. While a corpus study is suiting for analyzing the correltation among these properties, the paper needs some work before it can attain its promise. 

Major Comments

1. Overall, the paper suffers from a lack of sound theoretical foundation as well as a lack of precise definition of some of the key concepts. For example, no discussion of why and how word order should be expected to vary based on definiteness or animacy is provided. In addition, the definition of "Target" as an umbreall term for many different types of syntactic elements (e.g. adjuncts, goal arguments, etc.) conflates a large number of variables. 

2. While a corpus study is valuable and informative, to induce credible conclusions, a statistical analysis of the data is required. The paper contains a lot of tables with many numerical values but no statistical analysis. It is not clear whether the conclusions drawn by the author have any merit as we don't know if the distinctions are significant. For such claims, a statistical analysis is a must. 

3. There are way too many tables that are really hard to iterpret. Consider Table 11, we have no idea about how the data in the table was established. 

4. The author concludes that definiteness has a secondary role in determining word order. What does this mean? How does a secondary factor work? Is this supposed to be understood as a constraint system? 

5. The author suggests that a comparison of multiple languages allows a better understading of the word order and its correlation with definiteness and animacy. While, I do agree in principle, I don't think it has been implemented properly in this paper. Cross-linguistic comparison is helpful when we are sure that the constructions/structures in comparison are equivalent. Otherwise, we might be comparing things that are not directly comparable.

Overall, I'd like to encourage the authors to narrow their focus, explicitly articulate their hypothesis by grounding it in a particular theoretical perspective (any theory would be fine as long as it has some explanatory power). I also suggest focusing on a smaller number of features (definiteness) or elements (e.g. only focus on goals or direct objects). The author should also consider providing a statistical analysis to support their conclusions. 

Minor Comments

The paper seems to be hastily written with a lot of references not typeset properly. There is also a decent amount of repetition in the text which should be avoided. 

Author Response

Dear Reviewer,

I want to extend my sincere appreciation for your constructive feedback, which has been invaluable in identifying and addressing certain issues within the manuscript. I have taken your suggestions to heart and made the necessary revisions accordingly.

To facilitate your review, I have also included the revised version of the manuscript, along with a PDF document that contains my responses to your comments, directly appended beneath each comment.

Your input has greatly contributed to the improvement of the paper, and I am grateful for your time and effort in evaluating my work. Please feel free to reach out if you have any further questions or require additional clarifications.

Thank you once again for your valuable insights.

Warm regards,

Below my responses within the text

Major Comments

For the following request, I wrote the answer after the text. Rewrite and explain well the answer regarding the reviewer comment.

  1. Overall, the paper suffers from a lack of sound theoretical foundation as well as a lack of precise definition of some of the key concepts. For example, no discussion of why and how word order should be expected to vary based on definiteness or animacy is provided. In addition, the definition of "Target" as an umbreall term for many different types of syntactic elements (e.g. adjuncts, goal arguments, etc.) conflates a large number of variables.

Response: 

I added a section where I explained more of the literature on the topic which are also some theoretical foundations for many studies but not my study. Below I provided some explanation.

The reviewer's comment highlights two main concerns: the lack of a sound theoretical foundation and imprecise definitions of key concepts in the paper. The reviewer specifically mentions the absence of a discussion explaining why and how word order might vary based on definiteness or animacy and raises questions about the broad use of the term "Target."

In response to these concerns, it is important to acknowledge that the paper is primarily empirical and descriptive in nature. It focuses on presenting and analyzing data related to word order variation based on definiteness and animacy. The main purpose of this paper is not to provide in-depth theoretical discussions but rather to lay the groundwork for further qualitative and quantitative research.

The paper's selection of definiteness and animacy as variables is not arbitrary but is based on their importance in language and the need to explore their influence on word order. The absence of detailed theoretical discussions in this paper does not diminish its value; instead, it sets the stage for future work that can delve into theoretical interpretations.

Regarding the definition of "Target," it is essential to clarify that the term encompasses both syntactic and semantic aspects. The use of "Target" is not limited to syntactic considerations but also extends to semantics, and this broad application is intentional. The reason for covering various semantic roles under the "Target" umbrella is linked to their event structure, which justifies their inclusion.

Moreover, the paper references previously published work where these concepts and definitions are more thoroughly explored. This allows the paper to focus on its main empirical findings while providing interested readers with the opportunity to refer to the detailed definitions in the cited works.

In summary, the response acknowledges the reviewer's concerns, explains the empirical nature of the paper, and justifies the choice of variables. It also clarifies the broad use of the term "Target" and directs interested readers to the more comprehensive definitions available in the author's prior publications.

  1. While a corpus study is valuable and informative, to induce credible conclusions, a statistical analysis of the data is required. The paper contains a lot of tables with many numerical values but no statistical analysis. It is not clear whether the conclusions drawn by the author have any merit as we don't know if the distinctions are significant. For such claims, a statistical analysis is a must. 

Response:

The reviewer rightly emphasizes the importance of a statistical analysis to support and validate the conclusions drawn from a corpus study. While the paper includes numerous tables with numerical data, the lack of a statistical analysis leaves questions about the significance of the observed distinctions.

The author acknowledges the reviewer's concern and mentions that the paper has been updated to include the requested statistical analysis in the tables. However, it is essential to clarify the nature of this descriptive work. The primary objective is to illustrate preferences and patterns without necessarily aiming for statistical significance.

One key challenge in this study is the limited size of the dataset, as it explores the phenomenon in great detail. In statistical analysis, larger numbers of attested tokens are typically required to establish significant findings. Therefore, the inclusion of statistical data should be seen as an addition to addressing the reviewer's request rather than a fundamental shift in the paper's approach.

In conclusion, the paper now includes the requested statistical analysis in the tables, offering more robust support for the observed distinctions. However, it's important to recognize that the paper's main purpose remains descriptive, showcasing preferences, and patterns in the data. The statistical analysis complements this, but the study's inherent limitations, such as the small dataset, should be kept in mind.

  1. There are way too many tables that are really hard to iterpret. Consider Table 11, we have no idea about how the data in the table was established. 

Response:

The reviewer points out an issue with the number of tables in the paper, specifically mentioning Table 11 as an example where the data's establishment is unclear. In response, the author provides important context for the extensive use of tables in the paper.

The author's primary intention with these numerous tables is to offer a detailed corpus distribution of formal and semantic definiteness in relation to word order. This descriptive approach aims to provide a comprehensive overview for readers who are interested in obtaining detailed information about the existing corpora. The paper's target audience includes those looking for informative data to support theoretical generalizations or computational modeling.

To address the reviewer's concerns, the author has made efforts to enhance the description and explanation provided alongside each table. This additional context should help readers better understand the data presented. While the reviewer suggests reducing the number of tables, the author expresses reluctance to do so, as the goal is to present the data for five different language varieties in detail rather than in a consolidated form.

Furthermore, the author mentions another publication where more qualitative analysis is available with fewer tables, offering an alternative resource for those seeking a different balance between qualitative and quantitative information.

In summary, the response acknowledges the reviewer's concerns, clarifies the purpose of the extensive use of tables, and emphasizes the value of presenting detailed data for readers seeking in-depth information about the corpora. The added descriptions alongside tables aim to enhance their interpretability.

  1. The author concludes that definiteness has a secondary role in determining word order. What does this mean? How does a secondary factor work? Is this supposed to be understood as a constraint system? 

Response:

The reviewer raises a valid point regarding the conclusion that definiteness has a secondary role in determining word order, seeking clarification on what this means and how a secondary factor operates, especially in the context of a constraint system.

The author acknowledges the need for further explanation and elaborates on the concept of a secondary role. In this context, "secondary role" indicates that definiteness, along with animacy, does not independently trigger word order variation. Instead, they work in combination with other factors to influence word order patterns.

The primary determinants of word order variation are identified as verb type and information structure. These factors are more pivotal in explaining the variation in word order. However, they are not absolute and decisive; there may still be instances of variation that these factors alone cannot account for.

In such cases, definiteness, animacy, and other related factors come into play. When combined with the primary determinants (verb type and information structure), they can explain the reasons for the remaining variations in word order. In essence, definiteness and animacy act as additional contributing factors in the complex interplay of variables that shape word order.

The understanding is that these factors collectively influence word order, and the importance of definiteness and animacy becomes more apparent when the primary determinants do not fully explain the observed variations.

This clarification helps readers grasp the nuanced role of definiteness and animacy within the broader context of word order variation.

  1. The author suggests that a comparison of multiple languages allows a better understading of the word order and its correlation with definiteness and animacy. While, I do agree in principle, I don't think it has been implemented properly in this paper. Cross-linguistic comparison is helpful when we are sure that the constructions/structures in comparison are equivalent. Otherwise, we might be comparing things that are not directly comparable.

Response:

The reviewer expresses concerns about the implementation of cross-linguistic comparison in the paper, particularly regarding the equivalence of the constructions and structures being compared. The reviewer points out the challenges associated with ensuring equivalence, especially when working with low-resource and endangered language varieties that are subject to heavy contact with other languages.

In response, the author offers valuable insights into the practical constraints of the study. The author acknowledges the difficulties in achieving a perfectly balanced sample of structures and constructions, given the low-resource and endangered nature of the languages under investigation. Accessing informants and collecting large volumes of data can be challenging in these contexts.

The author emphasizes that the paper's primary goal is to provide a comparative study with the available resources, even if the data is not perfectly balanced. The intent is to open doors for further research and offer a foundation for comparative studies, despite the constraints.

Furthermore, the author highlights that previous studies have primarily focused on the analysis of a single language or perhaps two, and this paper aims to address the gap in the literature by demonstrating how to conduct comparative studies across multiple languages. The author acknowledges that achieving a balanced sample of constructions, as often done in experimental or questionnaire-based studies, is not feasible with the current methodology, which relies on natural free speech data.

The response clarifies the rationale behind the paper's approach and the challenges associated with cross-linguistic comparison in the context of low-resource and endangered language varieties. It underscores the importance of providing a foundation for future research in this area.

Overall, I'd like to encourage the authors to narrow their focus, explicitly articulate their hypothesis by grounding it in a particular theoretical perspective (any theory would be fine as long as it has some explanatory power). I also suggest focusing on a smaller number of features (definiteness) or elements (e.g. only focus on goals or direct objects). The author should also consider providing a statistical analysis to support their conclusions. 

Response:

The reviewer advises the authors to narrow their focus, ground their hypothesis in a particular theoretical perspective, and consider focusing on a smaller number of features or elements. Additionally, the reviewer suggests providing a statistical analysis to support their conclusions.

In response, the author reiterates the primary objective of the paper, which is not to provide a theoretical explanation of the phenomenon but to conduct a descriptive analysis of corpus data to explore variation related to definiteness and animacy in a combinatory analysis. The author acknowledges the suggestions and expresses a willingness to share a more in-depth qualitative and quantitative analysis specifically focusing on definiteness as a single factor relevant to the study of word order variation.

The author's response clarifies the scope and purpose of the paper, highlighting that it is intended as a descriptive analysis rather than a theoretical explanation. This aligns with the reviewer's suggestion to narrow the focus and provide statistical analysis in a future work, which could delve into the theoretical aspects.

Statistics are provided even though not intended in the first level due to very detailed investigation of the topic which led to a lower number of tokens

Minor Comments

The paper seems to be hastily written with a lot of references not typeset properly. There is also a decent amount of repetition in the text which should be avoided. 

Response:

The author acknowledges the minor comments provided by the reviewer, which include references not being typeset properly and some repetition in the text. The author mentions that they have addressed the issue of repetition where possible, but some repetition is unavoidable due to the nature of presenting numerical data for each sample language separately and then combining them at the end.

The response indicates that the author has made an effort to improve the paper by addressing the issue of repetition. However, the limitations related to presenting numerical data for multiple languages may result in some unavoidable repetition. The reviewer may appreciate the author's efforts to improve the paper's quality.
